# Discovering root causal genes with high-throughput perturbations

**Eric V Strobl[1]\*, Eric Gamazon[2]**

[1]University of Pittsburgh, Pittsburgh, United States; [2]Vanderbilt University Medical Center, Nashville, United States

## eLife Assessment

This work provides an **important** framework for understanding the primary causes of disease. While the theoretical results rely on strong assumptions about the underlying causal mechanisms, the authors provide **solid** empirical evidence that the framework is robust to modest violations of these assumptions.

**\*For correspondence:**
evs98@pitt.edu

**Competing interest:** The authors declare that no competing interests exist.

**Abstract** Root causal gene expression levels – or *root causal genes* for short – correspond to the initial changes to gene expression that generate patient symptoms as a downstream effect. Identifying root causal genes is critical towards developing treatments that modify disease near its onset, but no existing algorithms attempt to identify root causal genes from data. RNA-sequencing (RNA-seq) data introduces challenges such as measurement error, high dimensionality and non-linearity that compromise accurate estimation of root causal effects even with state-of-the-art approaches. We therefore instead leverage Perturb-seq, or high-throughput perturbations with single-cell RNA-seq readout, to learn the causal order between the genes. We then transfer the causal order to bulk RNA-seq and identify root causal genes specific to a given patient for the first time using a novel statistic. Experiments demonstrate large improvements in performance. Applications to macular degeneration and multiple sclerosis also reveal root causal genes that lie on known pathogenic pathways, delineate patient subgroups and implicate a newly defined omnigenic root causal model.

## Introduction

*Root causes of disease* correspond to the most upstream causes of a diagnosis. Intuitively, only a few root causes should have strong causal effects on the diagnosis, just like a machine breaks down due to a few root causal problems. *Pathogenesis* refers to the causal cascade from root causes to the diagnosis. Genetic and non-genetic factors may act as root causes and affect gene expression as an intermediate step during pathogenesis. We introduce root causal gene expression levels – or *root causal genes* for short – that correspond to the initial changes to *gene expression* induced by genetic and non-genetic root causes that have large causal effects on a downstream diagnosis (*Figure 1a*). Root causal genes differ from core genes that directly cause the diagnosis and thus lie at the end, rather than at the beginning, of pathogenesis (*Boyle et al., 2017*). Root causal genes also generalize (the expression levels of) driver genes that only account for the effects of somatic mutations primarily in cancer (*Martínez-Jiménez et al., 2020*).

Treating root causal genes can modify disease pathogenesis in its entirety, whereas targeting other causes may only provide symptomatic relief. For example, mutations in Gaucher disease cause decreased expression of wild type beta-glucocerebrosidase, or the root causal gene (*Nagral, 2014*). We can give a patient blood transfusions to alleviate the fatigue and anemia associated with the disease, but we seek more definitive treatments like recombinant glucocerebrosidase that replaces

**Figure 1.** Root causes, root causal genes and root causal effects. (**a**) Toy example where a variable $E_2$ simultaneously models genetic and non-genetic root causes that jointly have a large causal effect on a diagnosis $Y$ through gene expression $\widetilde{X}$. $E_2$ first affects the gene expression level $\widetilde{X}_2$, or the root causal gene. The root causal gene then affects other downstream levels during pathogenesis, including the core (or direct causal) gene $\widetilde{X}_4$, to ultimately induce $Y$. (**b**) We hypothesize that the causal effects of most root causes are small, but a few exert large causal effects (red ellipse), in each patient with disease. As a result, the distribution of these *root causal effects* tends to be right skewed in disease.

the deficient enzyme. Enzyme replacement therapy alleviates the associated liver, bone and neurological abnormalities of Gaucher disease as a downstream effect. Identifying root causal genes is therefore critical for developing treatments that eliminate disease near its pathogenic onset.

The problem is further complicated by the existence of complex disease, where a patient may have multiple root causal genes that differ from other patients even within the same diagnostic category (*Cano-Gamez and Trynka, 2020*). Complex diseases often have an overwhelming number of causes but, just like a machine usually breaks down due to one or a few root causal problems, the root causal genes may only represent a small subset of the genes because the causal effects of only a few root causes are large (*Figure 1b*). We thus also seek to identify *patient-specific* root causal genes in order to classify patients into meaningful biological subgroups each hopefully dictated by only a small group of genes.

No existing method identifies root causal genes from data. Many algorithms focus on discovering associational or predictive relations, sometimes visually represented as gene regulatory networks (*Costa-Silva et al., 2017*; *Ellington et al., 2023*). Other methods even identify causal relations (*Friedman et al., 2000*; *Wang et al., 2023*; *Wen et al., 2023*; *Buschur et al., 2020*), but none pinpoint the *first* gene expression levels that ultimately generate the vast majority of pathogenesis. Simply learning a causal graph does not resolve the issue because causal graphs do not summarize the effects of *unobserved* root causes, such as unmeasured environmental changes or variants, that are needed to identify all root causal genes. We therefore define the Root Causal Strength (RCS) score to identify all root causal genes unique to each patient. We then design the Root Causal Strength using Perturbations (RCSP) algorithm that estimates RCS from bulk RNA-seq under minimal assumptions by integrating Perturb-seq, or high-throughput perturbation experiments using CRISPR-based technologies coupled with single-cell RNA-sequencing (*Dixit et al., 2016*; *Adamson et al., 2016*; *Datlinger et al., 2017*). Experiments demonstrate marked improvements in performance, when investigators have access to a large bulk RNA-seq dataset and a genome-wide Perturb-seq dataset from a cell line of a disease-relevant tissue. Finally, application of the algorithm to two complex diseases with disparate pathogeneses recovers an *omnigenic root causal model*, where a small set of root causal genes drive pathogenesis but impact many downstream genes within each patient. As a result, nearly all gene expression levels are correlated with the diagnosis at the population level.

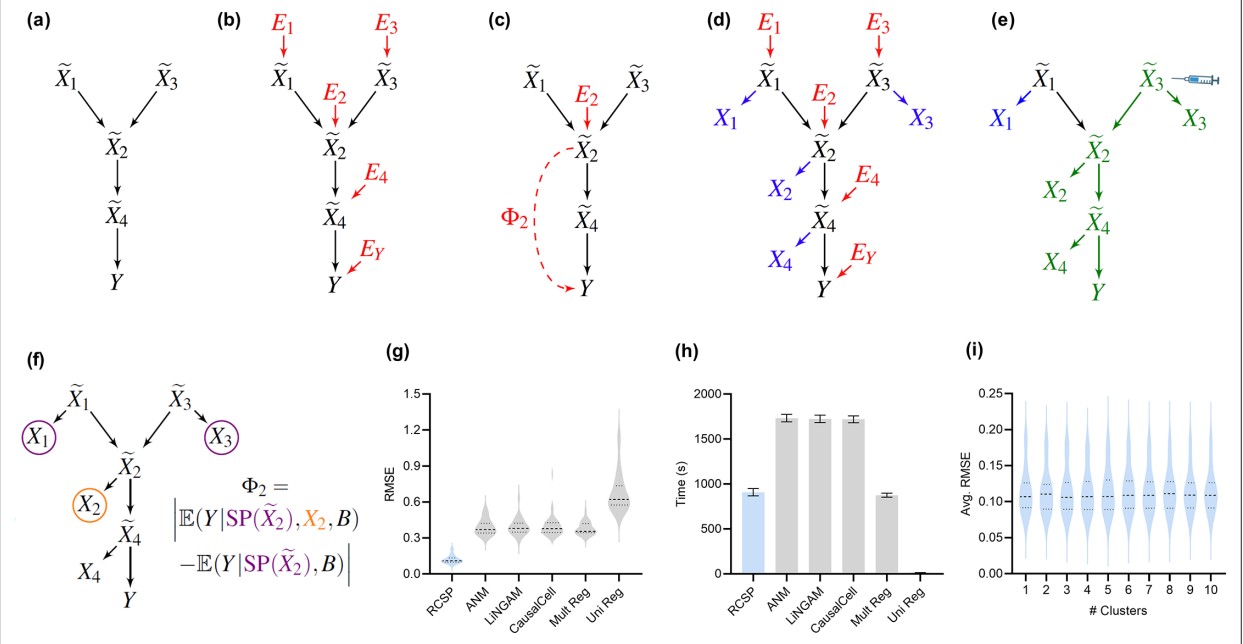

**Figure 2.** Method overview and synthetic data results. (**a**) We consider a latent causal graph over the true counts $\widetilde{X}$. (**b**) We augment the graph with error terms $E$ such that each $E_i \in E$ in red has an edge directed towards $\widetilde{X}_i \in \widetilde{X}$. (**c**) The RCS of $\widetilde{X}_2$, denoted by $\Phi_2$, quantifies the magnitude of the *conditional root causal effect*, or the strength of the causal effect from $E_2$ to $Y$ conditional on $\mathrm{Pa}(\widetilde{X}_2)$. (**d**) We cannot observe $\widetilde{X}$ in practice but instead observe the noisy surrogates $X$ in blue corrupted by Poisson measurement error. (**e**) Perturbing a variable such as $\widetilde{X}_3$ changes the marginal distributions of downstream variables shown in green under mild conditions. (**f**) RCSP thus uses the perturbation data to identify (an appropriate superset of) the surrogate parents for each variable in order to compute $\Phi$. (**g**) Violin plots show that RCSP achieved the smallest RMSE to the ground truth RCS values in the synthetic data. (**h**) RCSP also took about the same amount of time to complete as multivariate regression. Univariate regression only took 11 s on average, so its bar is not visible. Error bars denote 95% confidence intervals of the mean over 30 synthetic datasets. (**i**) Finally, RCSP maintained low RMSE values regardless of the number of clusters considered.

## Results

We briefly summarize the Methods in the first two subsections.

### Definitions

*Differential expression analysis* identifies differences in gene expression levels between groups $Y$ (*Costa-Silva et al., 2017*). A gene $X_i$ may be differentially expressed due to multiple reasons. For example, $X_i$ may cause $Y$, or a confounder $C$ may explain the relation between $X_i$ and $Y$ such that $X_i \leftarrow C \rightarrow Y$. In this paper, we take expression analysis a step further by pinpointing *causal* relations from expression levels regardless of the variable type of $Y$ (discrete or continuous). We in particular seek to discover *patient-specific root causal genes* from bulk RNA-seq data, which we carefully define below.

We represent a biological system in bulk RNA-seq as a causal graph $\mathbb{G}$ – such as in *Figure 2a* – where $p$ vertices $\widetilde{X}$ represent true gene expression levels in a bulk sample and $Y$ denotes the patient symptoms or diagnosis. The set $\widetilde{X}$ contains thousands of genes in practice. Directed edges between the vertices in $\mathbb{G}$ refer to direct causal relations. We assume that gene expression causes patient symptoms but not vice versa so that no edge from $Y$ is directed towards $\widetilde{X}$. The set $\mathrm{Pa}(\widetilde{X}_i)$ refers to the *parents* of $\widetilde{X}_i \in \widetilde{X}$, or those variables with an edge directed into $\widetilde{X}_i$. For example, $\mathrm{Pa}(\widetilde{X}_2) = \{\widetilde{X}_1, \widetilde{X}_3\}$ in *Figure 2a*. A *root vertex* corresponds to a vertex with no parents.

We can associate $\mathbb{G}$ with the structural equation $\widetilde{X}_i = f_i(\mathrm{Pa}(\widetilde{X}_i), E_i)$ for each $\widetilde{X}_i \in \widetilde{X}$ that links each vertex to its parents and error term $E_i$ (*Pearl, 2009*). The error term $E_i$ is not simply a regression residual but instead represents a combination of unobserved factors that only influence $\widetilde{X}_i$, such as unobserved transcriptional regulators, certain genetic variants and specific environmental conditions. We thus also include the error terms $E$ in the directed graph of *Figure 2b*. All root vertices are error terms and vice versa. The *root causes* of $Y$ are the error terms that cause $Y$, or have a directed path

into $Y$. We define the *root causal strength* (RCS) of $\widetilde{X}_i$ on $Y$ as the following absolute difference (*Figure 2c*):

$$
\begin{aligned}
\Phi_i &= \left| \mathbb{E}(Y|\text{Pa}(\widetilde{X}_i), E_i) - \mathbb{E}(Y|\text{Pa}(\widetilde{X}_i)) \right| \\
&= \left| \mathbb{E}(Y|\text{Pa}(\widetilde{X}_i), \widetilde{X}_i) - \mathbb{E}(Y|\text{Pa}(\widetilde{X}_i)) \right|.
\end{aligned}
\tag{1}
$$

We prove the last equality in the Materials and methods. As a result, RCS $\Phi_i$ directly measures the contribution of the gene $\widetilde{X}_i$ on $Y$ according to its error term $E_i$ without recovering the error term values. The algorithm does not impose distributional assumptions or functional restrictions such as additive noise to estimate the error term values as an intermediate step. Moreover $\Phi_i$ is patient-specific because the values of $\text{Pa}(\widetilde{X}_i)$ and $\widetilde{X}_i$ may differ between patients. We have $\Phi_i = 0$ when $E_i$ is not a cause of $Y$, and we say that the gene $\widetilde{X}_i$ is a *patient-specific root causal gene* if $\Phi_i \gg 0$, or its (conditional) root causal effect is large as depicted by the red ellipse in *Figure 1b*.

## Algorithm

We propose an algorithm called Root Causal Strength using Perturbations (RCSP) that estimates $\Phi = \{\Phi_1, \dots, \Phi_p\}$ from genes measured in both bulk RNA-seq and Perturb-seq datasets derived from possibly independent studies but from the same tissue type. We rely on bulk RNA-seq instead of single-cell RNA-seq in order to obtain many samples of the label $Y$. We focus on statistical estimation rather than statistical inference because $\Phi_i > 0$ when $E_i$ causes $Y$ under mild conditions, so we reject the null hypothesis that $\Phi_i = 0$ for many genes if many gene expression levels cause $Y$. However, just like a machine typically breaks down due to only one or a few root causal problems, we hypothesize that only a few genes have large RCS scores $\Phi_i \gg 0$ even in complex disease.

Estimating $\Phi$ requires access to the true gene expression levels $\widetilde{X}$ and the removal of the effects of confounding. We first control for batch effects representing unwanted sources of technical variation such as different sequencing platforms or protocols. We however can only obtain imperfect counts $X$ from RNA sequencing even within each batch (*Figure 2d*). Measurement error introduces confounding as well because it prevents us from exactly controlling for the causal effects of the gene expression levels. Investigators usually mitigate measurement error by normalizing the gene expression levels by sequencing depth. We show in the Materials and methods that the Poisson distribution approximates the measurement error distribution induced by the sequencing process to high accuracy (*Choudhary and Satija, 2022*; *Sarkar and Stephens, 2021*). We leverage this fact to eliminate the need for normalization by sequencing depth using an asymptotic argument where the library size $N$ approaches infinity. $N$ takes on a value of at least ten million in bulk RNA-seq, but we also empirically verify that the theoretical results hold well in the Appendix. We thus eliminate the Poisson measurement error and batch effects by controlling for the batches $B$ but not $N$ in non-linear regression models.

We in particular show that $\Phi_i$ in *Equation 1* is also equivalent to:

$$
\Phi_i = \left| \mathbb{E}(Y|\text{SP}(\widetilde{X}_i), X_i, B) - \mathbb{E}(Y|\text{SP}(\widetilde{X}_i), B) \right|,
\tag{2}
$$

where $\text{SP}(\widetilde{X}_i)$ refers to the *surrogate parents* of $\widetilde{X}_i$, or the variables in $X$ associated with $\text{Pa}(\widetilde{X}_i) \subseteq \widetilde{X}$. RCSP can identify (an appropriate superset of) the surrogate parents of each variable using perturbation data because perturbing a gene changes the marginal distributions of its downstream effects – which the algorithm detects from data under mild assumptions (*Figure 2e and f*). The algorithm thus only transfers the binary presence or absence of causal relations from the single cell to bulk data – rather than the exact functional relationships – in order to remain robust against discrepancies between the two data types; we empirically verify the robustness in the Appendix. RCSP finally performs the two non-linear regressions needed to estimate $\mathbb{E}(Y|\text{SP}(\widetilde{X}_i), X_i, B)$ and $\mathbb{E}(Y|\text{SP}(\widetilde{X}_i), B)$ for each $\Phi_i$. We will compare $\Phi_i$ against Statistical Dependence (SD), a measure of correlational strength defined as $\Omega_i = \left| \mathbb{E}(Y|X_i, B) - \mathbb{E}(Y|B) \right|$ where we have removed the conditioning on $\text{SP}(\widetilde{X}_i)$.

## In silico identification of root causal genes

We simulated 30 bulk RNA-seq and Perturb-seq datasets from random directed graphs summarizing causal relations between gene expression levels. We performed single gene knock-down perturbations

over 2500 genes and 100 batches. We obtained 200 cell samples from each perturbation, and another 200 controls without perturbations. We therefore generated a total of 2501×200=500, 200 single cell samples for each Perturb-seq dataset. We simulated 200 bulk RNA-seq samples. We compared RCSP against the Additive Noise Model (ANM; *Peters, 2014*; *Strobl and Lasko, 2023a*), the Linear Non-Gaussian Acyclic Model (LiNGAM; *Peters, 2014*; *Strobl and Lasko, 2022b*), CausalCell (*Wen et al., 2023*), univariate regression residuals (Uni Reg), and multivariate regression residuals (Multi Reg). The first two algorithms are state-of-the-art approaches used for error term extraction and, in theory, root causal discovery. See Materials and methods for comprehensive descriptions of the simulation setup and comparator algorithms.

We summarize accuracy results in *Figure 2g* using the Root Mean Squared Error (RMSE) to the ground truth Φ values. All statements about pairwise differences hold true at a Bonferonni corrected threshold of 0.05/5 according to paired two-sided t-tests, since we compared RCSP against a total of five algorithms. RCSP estimated Φ most accurately by a large margin. ANM and LiNGAM are theoretically correct under their respective assumptions, but they struggle to outperform standard multivariate regression due to the presence of measurement error in RNA-seq (Appendix). Feature selection and causal discovery with CausalCell did not improve performance. Univariate regression performed the worst, since it does not consider the interactions between variables. RCSP achieved the lowest RMSE while completing in about the same amount of time as multivariate regression on average (*Figure 2h*). RCSP maintained the lowest RMSE even in the cyclic case, and the performance of the algorithm remained robust to differences between the directed graphs underlying the bulk RNA-seq and Perturb-seq data (Appendix). We conclude that RCSP both scalably and accurately estimates Φ.

We will cluster the RCS values in real data to find patient subgroups. We therefore also performed hierarchical clustering using Ward's method (*Ward, 1963*) on the values of Φ estimated by RCSP with the synthetic data. We then computed the RMSEs and averaged them within each cluster. We found that RCSP maintained low average RMSE values regardless of the number of clusters considered (*Figure 2i*). We conclude that RCSP maintains accurate estimation of Φ across different numbers of clusters.

## Oxidative stress in age-related macular degeneration

We ran RCSP on a bulk RNA-seq dataset of 513 individuals with age-related macular degeneration (AMD; GSE115828) and a Perturb-seq dataset of 247,914 cells generated from an immortalized retinal pigment epithelial (RPE) cell line (*Ratnapriya et al., 2019*; *Replogle et al., 2022*). The Perturb-seq dataset contains knockdown experiments of 2077 genes overlapping with the genes of the bulk dataset. We set the target $Y$ to the Minnesota Grading System score, a measure of the severity of AMD based on stereoscopic color fundus photographs. We always included age and sex as a biological variable as covariates. We do not have access to the ground truth values of Φ in real data, so we evaluated RCSP using seven alternative techniques. See Materials and methods for a detailed rationale of the evaluation of real data. RCSP outperformed all other algorithms in this dataset (Appendix). We therefore only analyze the output of RCSP in detail here.

AMD is a neurodegenerative disease of the aging retina (*Hadziahmetovic and Malek, 2020*), so age is a known root cause of the disease. We therefore determined if RCSP identified age as a root cause. Note that RCSP does not need perturbation data of age to compute the RCS values of age, since age has no parents in the directed graph. The algorithm estimated a heavy tailed distribution of the RCS values indicating that most of the RCS values deviated away from zero (*Figure 3a*). The Deviation of the RCS (D-RCS), or the deviation from an RCS value of zero, measures the tailedness of the distribution while preserving the unit of measurement. The D-RCS of age corresponded to 0.46 – more than double that of the nearest gene (*Figure 3d*). We conclude that RCSP correctly detected age as a root cause of AMD.

Root causal genes typically affect many downstream genes before affecting $Y$. We therefore expect to identify few root causal genes but many genes that correlate with $Y$. To evaluate this hypothesis, we examined the distribution of D-RCS relative to the distribution of the Deviation of Statistical Dependence (D-SD), or the deviation from an SD value of zero, in *Figure 3b*. Notice that the histogram of D-RCS scores in *Figure 3b* mimics a folded distribution of *Figure 1b*. Thus, few D-RCS scores had large values implying the existence of only a few root causal genes. In contrast, most of the D-SD scores had relatively larger values concentrated around 0.10 implying the existence of many

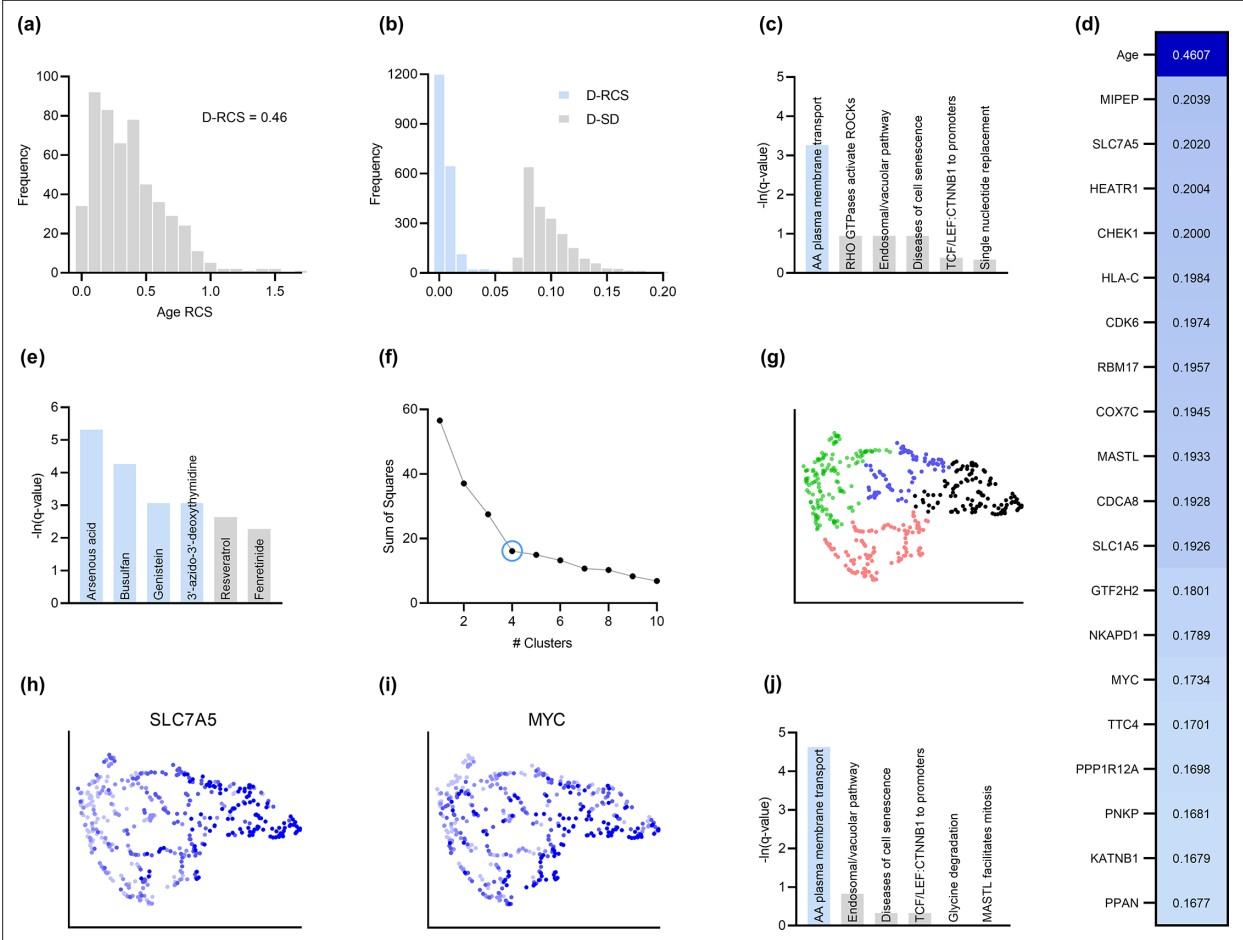

**Figure 3.** Analysis of AMD. (**a**) The distribution of the RCS scores of age deviated away from zero and had a composite D-RCS of 0.46. (**b**) However, the majority of gene D-RCS scores concentrated around zero, whereas the majority of gene D-SD scores concentrated around the relatively larger value of 0.10. Furthermore, the D-RCS scores of the genes in (**d**) mapped onto the 'amino acid transport across the plasma membrane' pathway known to be involved in the pathogenesis of AMD in (**c**). Blue bars survived 5% FDR correction. (**e**) Drug enrichment analysis revealed four significant drugs, the later three of which have therapeutic potential. (**f**) Hierarchical clustering revealed four clear clusters according to the elbow method, which we plot by UMAP dimensionality reduction in (**g**). The RCS scores of the top genes in (**d**) increased only from the left to right on the first UMAP dimension (x-axis); we provide an example of SLC7A5 in (**h**) and one of three detected exceptions in (**i**). We therefore performed pathway enrichment analysis on the black cluster in (**g**) containing the largest RCS scores. (**j**) The amino acid transport pathway had a larger degree of enrichment in the black cluster as compared to the global analysis in (**c**).

genes correlated with $Y$. We conclude that RCSP identified few root causal genes rather than many correlated genes for AMD.

The pathogenesis of AMD involves the loss of RPE cells. The RPE absorbs light in the back of the retina, but the combination of light and oxygen induces oxidative stress, and then a cascade of events such as immune cell activation, cellular senescence, drusen accumulation, neovascularization and ultimately fibrosis (*Barouch and Miller, 2007*). We therefore expect the root causal genes of AMD to include genes involved in oxidative stress during early pathogenesis. The gene MIPEP with the highest D-RCS score in *Figure 3d* indeed promotes the maturation of oxidative phosphorylation-related proteins (*Shi et al., 2011*). The second gene SLC7A5 is a solute carrier that activates mTORC1 whose hyperactivation increases oxidative stress via lipid peroxidation (*Nachef et al., 2021*; *Go et al., 2020*). The gene HEATR1 is involved in ribosome biogenesis that is downregulated by oxidative stress (*Turi et al., 2018*). The top genes discovered by RCSP thus identify pathways known to be involved in oxidative stress. We further verified that measurement error did not explain their large D-RCS scores in the Appendix.

We subsequently jointly analyzed the D-RCS values of all 2077 genes. We performed pathway enrichment analysis that yielded one pathway 'amino acid transport across the plasma membrane' that passed an FDR threshold of 5% (*Figure 3c*). The leading edge genes of the pathway included the solute carriers SLC7A5 and SLC1A5. These two genes function in conjunction to increase the efflux of essential amino acids out of the lysosome (*Nicklin et al., 2009*; *Beaumatin et al., 2019*). Some of these essential amino acids like L-leucine and L-arginine activate mTORC1 that in turn increases lipid peroxidation induced oxidative stress and the subsequent degeneration of the RPE (*Nachef et al., 2021*; *Go et al., 2020*). We conclude that pathway enrichment analysis correctly identified solute carrier genes involved in a known pathway promoting oxidative stress in AMD.

We next ran drug enrichment analysis with the D-RCS scores. The top compound arsenous acid inhibits RPE proliferation (*Su et al., 2020*), but the other three significant drugs have therapeutic potential (*Figure 3e*). Busulfan decreases the requirement for intravitreal anti-VEGF injections (*Dalvin et al., 2022*). Genistein is a protein kinase inhibitor that similarly attenuates neovascularization (*Kinoshita et al., 2014*) and blunts the effect of ischemia on the retina (*Kamalden et al., 2011*). Finally, a metabolite of the antiviral agent 3'-azido-3'-deoxythymidine inhibits neovascularization and mitigates RPE degeneration (*Narendran et al., 2020*). We conclude that the D-RCS scores identified promising drugs for the treatment of AMD.

Hierarchical clustering and UMAP dimensionality reduction on the patient-specific RCS values revealed four clear clusters of patients by the elbow method on the sum of squares plot (*Figure 3f and g*, respectively). The RCS scores of most of the top genes exhibited a clear gradation increasing only from the left to the right hand side of the UMAP embedding; we plot an example in *Figure 3h*. We found three exceptions to this rule among the top 30 genes (example in *Figure 3i* and see the Appendix). RCSP thus detected genes with large RCS scores primarily in the black cluster of *Figure 3g*. Pathway enrichment analysis within this cluster alone yielded supra-significant results on the same pathway detected in the global analysis (*Figure 3j* versus *Figure 3c*). Furthermore, drug enrichment analysis results by cluster confirmed that patients in the black cluster with many root causal genes are likely the hardest to treat (Appendix). We conclude that RCSP detected a subgroup of patients whose root causal genes have large RCS scores and involve known pathogenic pathways related to oxidative stress.

## T cell infiltration in multiple sclerosis

We next ran RCSP on 137 samples collected from CD4+ T cells of multiple sclerosis (MS; GSE137143) as well as Perturb-seq data of 1,989,578 K562 cells, which can be genetically engineered into artificial antigen-presenting cells for expanding T cells (*Butler and Hirano, 2014*; *Replogle et al., 2022*). We set the target $Y$ to the Expanded Disability Status Scale score, a measure of MS severity. RCSP outperformed all other algorithms in this dataset as well (Appendix).

MS progresses over time, and RCSP correctly detected age as a root cause of MS severity with RCS values deviating away from zero (*Figure 4a*). The distribution of gene D-RCS scores concentrated around zero with a long tail, whereas the distribution of gene D-SD scores concentrated around a relatively larger value of 0.3 (*Figure 4b*). RCSP thus detected an omnigenic root causal model with a few root causal genes but many correlated genes.

MS is an inflammatory neurodegenerative disease that damages the myelin sheaths of nerve cells in the brain and spinal cord. T cells may mediate the inflammatory process by crossing a disrupted blood brain barrier and repeatedly attacking the myelin sheaths (*Fletcher et al., 2010*). Damage induced by the T cells also perturbs cellular homeostasis and leads to the accumulation of misfolded proteins (*Andhavarapu et al., 2019*). The root causal genes of MS thus likely include genes involved in T cell infiltration across the blood brain barrier.

Genes with the highest D-RCS scores included MNT, CERCAM, and HERPUD2 (*Figure 4d*). MNT is a MYC antagonist that modulates the proliferative and pro-survival signals of T cells after engagement of the T cell receptor (*Gnanaprakasam and Wang, 2017*). Similarly, CERCAM is an adhesion molecule expressed at high levels in microvessels of the brain that increases leukocyte transmigration across the blood brain barrier (*Starzyk et al., 2000*). HERPUD2 is involved in the endoplasmic-reticulum associated degradation of unfolded proteins (*Kokame et al., 2000*). Genes with the highest D-RCS scores thus serve key roles in known pathogenic pathways of MS.

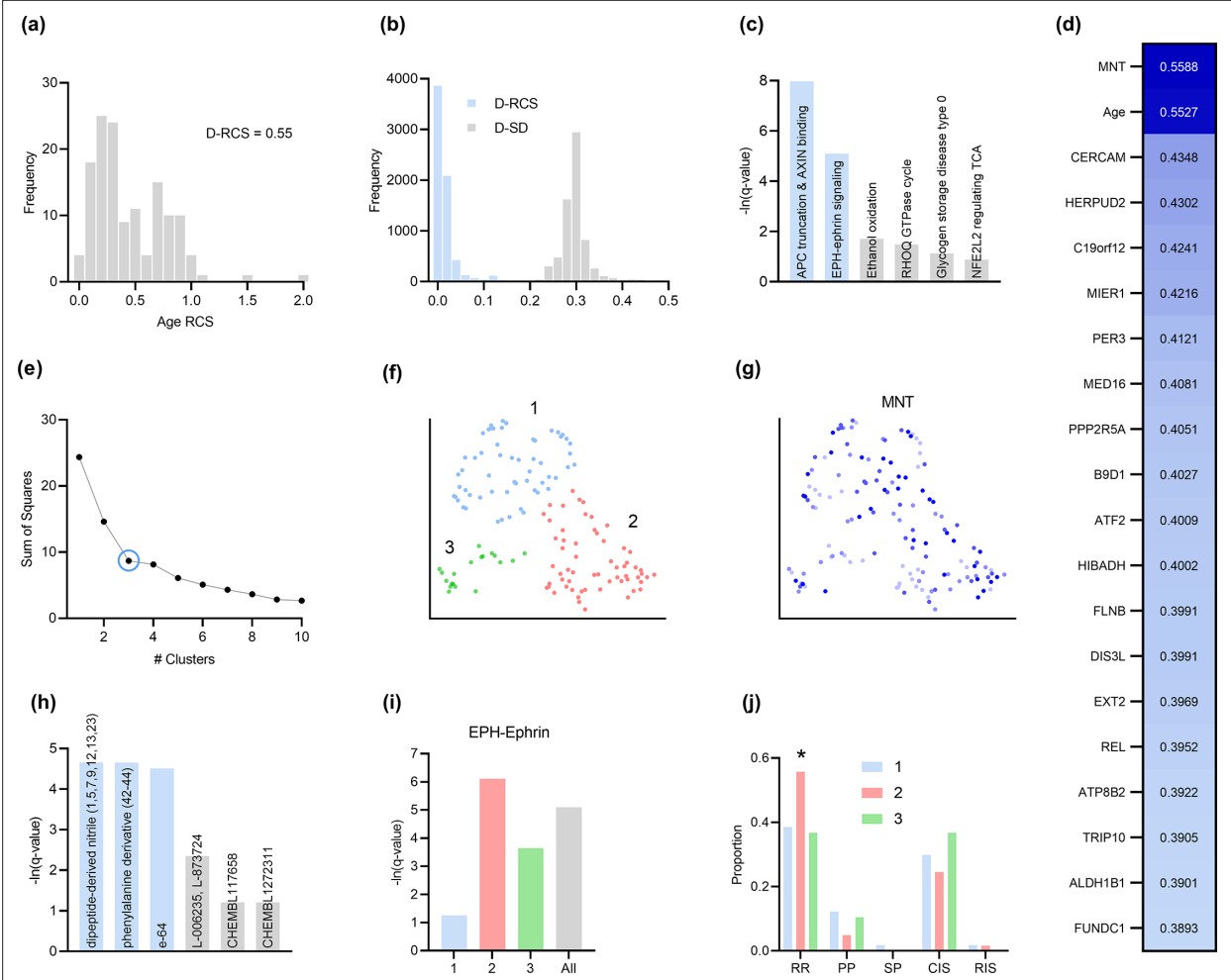

**Figure 4.** Analysis of MS. (**a**) The distribution of the RCS scores of age deviated away from zero with a composite D-RCS of 0.55. (**b**) The distribution of D-RCS concentrated around zero, whereas the distribution of D-SD concentrated around 0.3. (**d**) RCSP identified many genes with large D-RCS scores that in turn mapped onto known pathogenic pathways in MS in (**c**). Hierarchical clustering revealed three clusters in (**e**), which we plot in two dimensions with UMAP in (**f**). Top genes did not correlate with either dimension of the UMAP embedding; we provide an example of the MNT gene in (**g**). (**h**) Drug enrichment analysis in the green cluster implicated multiple cathepsin inhibitors. Finally, EPH-ephrin signaling survived FDR correction in (**c**) and was enriched in the pink cluster in (**i**) which contained more MS patients with the relapsing-remitting subtype in (**j**); subtypes include relapsing-remitting (RR), primary progressive (PP), secondary progressive (SP), clinically isolated syndrome (CIS), and radiologically isolated syndrome (RIS).

We found multiple genes with high D-RCS scores in MS, in contrast to AMD where age dominated (*Figure 4d* versus *Figure 3d*). Measurement error did not account for the high scores (Appendix). We performed pathway enrichment analysis using the D-RCS scores of all genes and discovered two significant pathways at an FDR corrected threshold of 5%: 'adenomatous polyposis coli (APC) truncation mutants have impaired AXIN binding' and 'EPH-ephrin signaling' (*Figure 4c*). APC and AXIN are both members of the Wnt signaling pathway and regulate levels of beta-catenin (*Spink et al., 2000*). Furthermore, inhibition of Wnt/beta-catenin causes CD4+ T cell infiltration into the central nervous system via the blood brain barrier in MS (*Lengfeld et al., 2017*). Ephrins similarly regulate T cell migration into the central nervous system (*Luo et al., 2016*) and are overexpressed in MS lesions (*Sobel, 2005*). The APC-AXIN and EPH-ephrin pathways are thus consistent with the known pathophysiology of central nervous system T cell infiltration in MS.

We subsequently performed hierarchical clustering of the RCS scores. The within cluster sum of squares plot in *Figure 4e* revealed the presence of three clusters by the elbow method. We plot the three clusters in a UMAP embedding in *Figure 4f*. The clusters did not show a clear relationship with MS symptom severity (Appendix) or the levels of the top most genes of *Figure 4d*; we plot the MNT gene as an example in *Figure 4g*. However, further analyses with additional genes revealed that the

distribution of many lower ranked genes governed the structure of the UMAP embedding (Appendix). The D-RCS scores of each cluster also implicated different mechanisms of T cell pathology including APC-AXIN in the green cluster, disturbed T cell homeostasis in the pink cluster and platelet enhanced T cell autoreactivity in the blue cluster (Appendix).

Global drug enrichment analysis did not yield any significant drugs even at a liberal FDR threshold of 10%. We thus ran drug enrichment analysis in each cluster of *Figure 4f*. The blue and pink clusters again did not yield significant drugs. However, the third green cluster identified the cysteine cathepsin inhibitors dipeptide-derived nitriles, phenylalinine derivatives, e-64, L-006235 and L-873724 (*Figure 4h*); statistical significance of the first three held even after correcting for multiple comparisons with the Bonferroni adjustment of 0.05/4 on the q-values. The leading edge genes of the significant drugs included the cathepsins CTSL, CTSS, and CTSB exclusively. These drug enrichment results corroborate multiple experimental findings highlighting the therapeutic efficacy of cathepsin inhibitors in a subgroup of MS patients responsive to interferon therapy (*Haves-Zburof et al., 2011*; *Burster et al., 2007*).

Prior research has also shown that EPH-ephrin signaling is more prevalent in relapsing-remitting multiple sclerosis than in other subtypes of the disease (*Golan et al., 2021*). EPH-ephrin signaling survived FDR correction in our analysis (*Figure 4c*). Furthermore, the pathway was more enriched in the pink cluster than in the other two (*Figure 4i*). The pink cluster indeed contained a higher proportion of patients with the relapsing-remitting subtype (*Figure 4j*). RCSP thus precisely identified the enrichment of EPH-ephrin signaling in the correct subtype of MS.

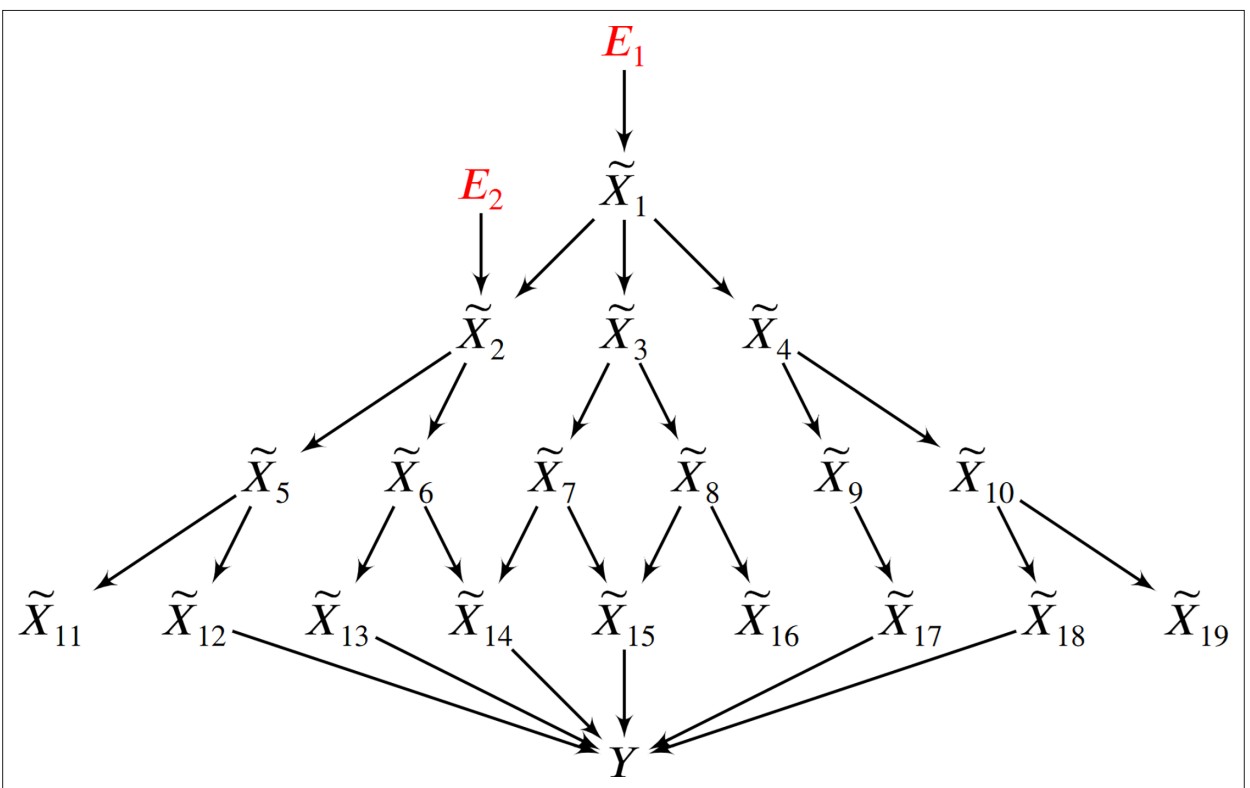

**Figure 5.** In this example, two root causal genes $\widetilde{X}_1$ and $\widetilde{X}_2$ affect many downstream genes and ultimately cause $Y$. Thus all genes $\widetilde{X}_1, \ldots, \widetilde{X}_{19}$ correlate with $Y$, but only $\widetilde{X}_1$ and $\widetilde{X}_2$ have large root causal effects on $Y$. The omnigenic root causal model posits that only a few root causal genes affect many downstream genes, so that nearly all genes are correlated with $Y$. Causal genetic variants can directly cause $Y$ or cause any gene expression level that causes $Y$ – including those with small root causal effects – but only $\widetilde{X}_1$ and $\widetilde{X}_2$ have large root causal effects on $Y$ due to genetic *and non-genetic* root causes modeled by $E_1$ and $E_2$. In contrast, the core gene model assumes only a few direct causal genes $\widetilde{X}_{12}, \widetilde{X}_{13}, \widetilde{X}_{14}, \widetilde{X}_{15}, \widetilde{X}_{17}, \widetilde{X}_{18}$. These core genes do not account for the deleterious causal effects of $E_1$ and $E_2$ on $\widetilde{X}_{11}, \widetilde{X}_{16}$ and $\widetilde{X}_{19}$.

## Discussion

We presented a framework for identifying root causal genes, or the gene expression levels directly regulated by root causes with large causal effects on $Y$, by modeling the root causes using the error terms of structural equation models. Each error term represents the conglomeration of unobserved root causes, such as genetic variants or environmental conditions, that directly cause a specific gene. We however do not have access to many of the error terms in practice, so we introduced the root causal strength (RCS) score, or the magnitude of the conditional causal effect of each error term, which we can compute using gene expression levels alone. The RCSP algorithm computes RCS given knowledge of the causal ancestors of each variable, which we obtained by Perturb-seq. RCSP only transfers the causal structure (binary cause-effect relations) from the single cell to bulk data rather than the exact functional relationships in order to remain robust against discrepancies between the two data types. Results with the synthetic data demonstrated marked improvements over existing alternatives. The algorithm also recovered only a few root causal genes that play key roles in known pathogenic pathways and implicate therapeutic drugs in both AMD and MS.

We detected a modest number of root causal genes in both AMD and MS, but virtually all genes were correlated with $Y$. This omnigenic model, where 'omni-' refers to the nearly all genes correlated with $Y$, differs from the omnigenic model involving *core genes* (*Boyle et al., 2017*). Boyle et al. define core genes as genes that directly affect disease risk. The authors further elaborate that many *peripheral genes* affect the functions of a modest number of core genes, so the peripheral genes often explain most of disease heritability. In contrast, root causal genes may not directly cause $Y$ but lie substantially upstream of $Y$ in the causal graph. The error terms of upstream root causal genes affect many downstream genes that include both ancestors and non-ancestors of $Y$ (*Figure 5*). These downstream genes contain traces of the root causal gene error terms that induce the many correlations with $Y$. The root causal model thus assumes sparsity in upstream root causal genes, whereas the core gene model assumes sparsity in the downstream direct causal genes; the omnigenic root causal model makes no statement about the number of direct causal genes, so direct causal genes may be sparse or dense. Further, each causal genetic variant tends to have only a small effect on disease risk in complex disease because the variant can directly cause $Y$ or directly cause any causal gene including those with small root causal effects on $Y$; thus, all error terms that cause $Y$ can model genetic effects on $Y$. However, the root causal model further elaborates that genetic *and non-genetic factors* often combine to produce a few root causal genes with large root causal effects, where non-genetic factors typically account for the majority of the large effects in complex disease. Many variants may therefore cause many genes in diseases with only a few root causal genes. We finally emphasize that the root causal model accounts for all deleterious effects of the root causal genes, whereas the core gene model only captures the deleterious effects captured by the diagnosis $Y$. For example, the *disease* of diabetes causes retinopathy, but retinopathy is not a part of the diagnostic criteria of diabetes. As a result, the gene expression levels that cause retinopathy but not the *diagnosis* of diabetes are not core genes, even though they are affected by the root causal genes. The sparsity of the root causal genes, the focus on the combined effects of genetic and non-genetic root causes, and the ability to account for root causal effects not represented by the target $Y$ motivate us to use the phrase *omnigenic root causal model* in order to distinguish it from the omnigenic core gene model.

We identified root causal genes without imposing parametric assumptions using the RCS metric. Prior measures of root causal effect require restrictive functional relations, such as linear relations or additive noise, and continuous random variables (*Strobl and Lasko, 2022b*; *Strobl et al., 2024*; *Strobl and Lasko, 2023a*). These restrictions ensure exact identifiability of the underlying causal graph and error terms. However, real RNA-seq is obtained from a noisy sequencing process and contains count data arguably corrupted by Poisson measurement error (*Sarkar and Stephens, 2021*). The Poisson measurement error introduces confounding that precludes exact recovery of the underlying error terms. The one existing root causal discovery method that can handle Poisson measurement error uses single-cell RNA-seq, estimates negative binomial distribution parameters and cannot scale to the thousands of genes required for meaningful root causal detection (*Strobl and Lasko, 2023b*). RCSP rectifies the deficiencies of these past approaches by ensuring accurate root causal detection even in the presence of the counts, measurement error and high dimensionality of RNA-seq.

This study carries other limitations worthy of addressing in future work. The RCS score importantly quantifies root causal strength rather than root causal effect. As a result, the method cannot be used

to identify the direction of root causal effect unconditional on the parents. The root causal effect and signed RCS (or expected conditional root causal effect) do not differ by much in practice (Appendix), but future work may focus on exactly identifying both the strength and direction of the unconditional causal effects of the error terms. Furthermore, RCS achieves patient but not cell-type specificity because the algorithm relies on phenotypic labels obtained from bulk RNA-seq. RCSP thus cannot identify the potentially different root causal genes present within distinct cell populations. Modern genome-wide Perturb-seq datasets also adequately perturb and measure only a few thousand, rather than all, gene expression levels. RCSP can only identify root causal genes within this perturbed and measured subset. Fourth, RCSP accounts for known batch effects and measurement error but cannot adjust for unknown confounding. Finally, RCSP assumes a directed acyclic graph. We can transform a directed graph with cycles into an acyclic one under equilibrium, but real biological distributions vary across time (*Spirtes, 1995*; *Bongers et al., 2021*). Future work should thus aim to estimate cell-type specific root causal effects under latent confounding and time-varying distributions.

In conclusion, RCSP integrates bulk RNA-seq and Perturb-seq to identify patient-specific root causal genes under a principled causal inference framework using the RCS score. RCS quantifies root causal strength implicitly without requiring normalization by sequencing depth or direct access to the error terms of a structural equation model. The algorithm identifies the necessary causal relations to compute RCS using reliable high-throughput perturbation data rather than observational data alone. The RCS scores often suggest an omnigenic root causal model of disease. Enrichment analyses with the RCS scores frequently reveal pathogenic pathways and drug candidates. We conclude that RCSP is a novel, accurate, scalable and disease-agnostic procedure for performing patient-specific root causal gene discovery.

## Materials and methods
### Background on causal discovery
We denote a singleton variable like $\widetilde{X}_i$ with italics and sets of variables like $\widetilde{X}$ with bold italics. We can represent a causal process using a *structural equation model* (SEM) linking the $p + 1$ variables in $\boldsymbol{Z} = \widetilde{\boldsymbol{X}} \cup Y$ using a series of deterministic functions:

$$Z_i = f_i(\mathrm{Pa}(Z_i), E_i), \qquad \forall Z_i \in \boldsymbol{Z} \tag{3}$$

where $f_i$ is a function of the *parents*, or direct causes, of $Z_i$ and an error term $E_i \in \boldsymbol{E}$. The error terms $\boldsymbol{E}$ are mutually independent. We will use the terms *vertex* and *variable* interchangeably. A *root vertex* corresponds to a vertex without any parents. On the other hand, a *terminal* or *sink vertex* is not a parent of any other vertex.

We can associate a directed graph to $\boldsymbol{Z}$ by drawing a directed edge from each member of $\mathrm{Pa}(Z_i)$ to $Z_i$ for all $Z_i \in \boldsymbol{Z}$. A *directed path* from $Z_i$ to $Z_j$ corresponds to a sequence of adjacent directed edges from $Z_i$ to $Z_j$. If such a path exists (or $Z_i = Z_j$), then $Z_i$ is an *ancestor* of $Z_j$ and $Z_j$ is a *descendant* of $Z_i$. We collate all ancestors of $Z_i$ into the set $\mathrm{Anc}(Z_i)$. A *cycle* occurs when there exists a directed path from $Z_i$ to $Z_j$ and the directed edge $Z_j \to Z_i$. A *directed acyclic graph* (DAG) contains no cycles. We *augment* a directed graph by including additional vertices $\boldsymbol{E}$ and drawing a directed edge from each $E_i \in \boldsymbol{E}$ to $X_i$ except when $X_i = E_i$ is already a root vertex. We consider an augmented DAG $\mathbb{G}$ throughout the remainder of this manuscript.

The vertices $Z_i$ and $Z_j$ are *d-connected* given $\boldsymbol{W} \subseteq \boldsymbol{Z} \setminus \{Z_i, Z_j\}$ in $\mathbb{G}$ if there exists a path between $Z_i$ and $Z_j$ such that every collider on the path is an ancestor of $\boldsymbol{W}$ and no non-collider is in $\boldsymbol{W}$. The vertices are *d-separated* if they are not d-connected. Any DAG associated with the SEM in *Equation 3* also obeys the *global Markov property* where $Z_i$ and $Z_j$ are conditionally independent given $\boldsymbol{W}$ if they are d-separated given $\boldsymbol{W}$. The term *d-separation faithfulness* refers to the converse of the global Markov property where conditional independence implies d-separation. A distribution obeys *unconditional d-separation faithfulness* when we can only guarantee d-separation faithfulness when $\boldsymbol{W} = \emptyset$.

### Causal modeling of RNA sequencing
Performing causal discovery requires careful consideration of the underlying generative process. We therefore propose a causal model for RNA-seq. We differentiate between the biology and the RNA sequencing technology.

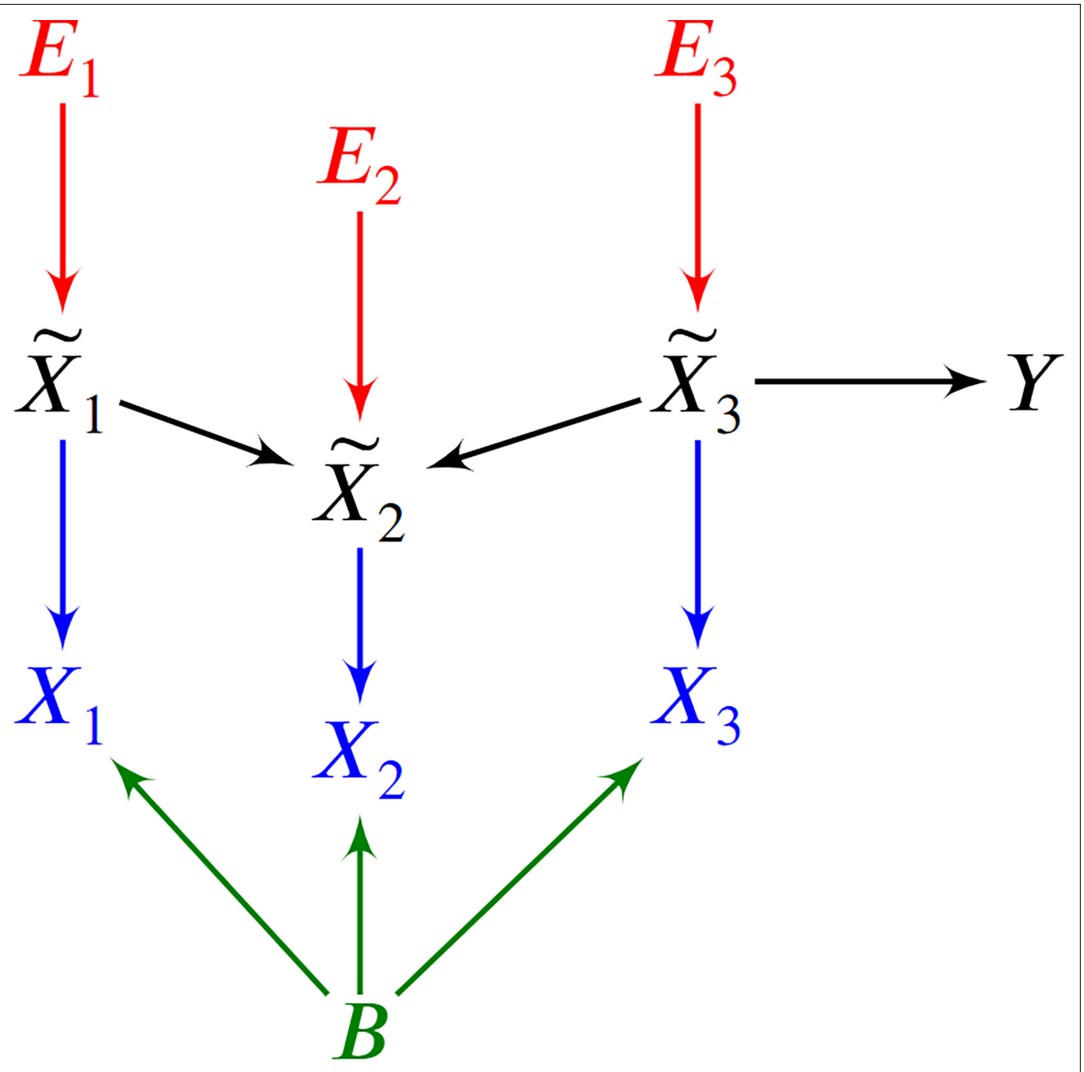

**Figure 6.** An example of a DAG over $X \cup \widetilde{X} \cup B \cup Y$ augmented with the error terms $E$. The observed vertices $X$ denote counts corrupted by batch $B$ effects and Poisson measurement error.

We represent a snapshot of a biological causal process using an SEM over $\widetilde{X} \cup Y$ obeying **Equation 3**. We assume that the phenotypic target $Y$ is a terminal vertex so that gene expression causes phenotype but not vice versa. Each $\widetilde{X}_i \in \widetilde{X}$ corresponds to the total number of RNA molecules of a unique gene in a single cell or bulk tissue sample. The error terms model root causes that are outside of gene expression, such as genetic variation or environmental factors. Moreover, the relation from gene expression to $Y$ is stochastic because $Y = f_Y(\text{Pa}(Y), E_Y)$, where $E_Y$ introduces the stochasticity. Two individuals may therefore have the exact same error term values over $\widetilde{X}$ but different instantiations of $Y$.

We unfortunately cannot observe $\widetilde{X}$ in practice but instead measure a corrupted count $X$ using single cell or bulk RNA-seq technology. We derive the measurement error distribution from first principles. We map an exceedingly small fraction of each $\widetilde{X}_i \in \widetilde{X}$ within a sample at unequal coverage. Let $\pi_{ij}$ denote the probability of mapping one molecule of $\widetilde{X}_i$ in batch $j$ so that $\sum_{i=1}^{p} \pi_{ij}$ is near zero. The law of rare events (**Papoulis, 1984**) implies that the Poisson distribution well-approximates the library size $N$ so that $N \sim \text{Pois}(\sum_{i=1}^{p} \widetilde{X}_i \pi_{ij})$.

We write the probability of mapping $\widetilde{X}_i$ in a given sample as:

$$P_{ij} = \frac{\widetilde{X}_i \pi_{ij}}{\sum_{i=1}^p \widetilde{X}_i \pi_{ij}}.$$

This proportion remains virtually unchanged when sampling without replacement because $N \ll \sum_{i=1}^p \widetilde{X}_i$ with small $\sum_{i=1}^p \pi_{ij}$. We can therefore approximate sampling *without* replacement by sampling *with* replacement using a multinomial: $X \sim \text{MN}(N; P_{1j}, \ldots, P_{pj})$. This multinomial and the Poisson distribution over $N$ together imply that the marginal distribution of each $X_i \in X$ follows an independent Poisson distribution centered at $(\sum_{i=1}^p \widetilde{X}_i \pi_{ij})P_{ij} = \widetilde{X}_i \pi_{ij}$, or:

$$X_i \sim \text{Pois}(\widetilde{X}_i \pi_{ij}). \tag{4}$$

We conclude that the measurement error distribution follows a Poisson distribution to high accuracy. Multiple experimental results already corroborate this theoretical conclusion (*Grün et al., 2014*; *Sarkar and Stephens, 2021*; *Choudhary and Satija, 2022*).

We can represent the biology and the RNA sequencing in a single DAG over $X \cup \widetilde{X} \cup B \cup Y$, where $B$ denotes the batch, and $Y$ the target variable representing patient symptoms or diagnosis. We provide a toy example in *Figure 6*. We draw $\mathbb{G}$ over $Z$ in black and make each $\widetilde{X}_i \in \widetilde{X}$ a parent of $X_i \in X$ in blue. We then include the root vertex $B$ as a parent of all members of $X$ in green. We augment this graph with the error terms of $\widetilde{X}$ in red and henceforth refer to the augmented DAG as $\mathbb{G}$. Repeated draws from the represented causal process generates a dataset.

## No need for normalization by sequencing depth

We provide an asymptotic argument that eliminates the need for normalization by sequencing depth when estimating conditional expectations using bulk RNA-seq. The argument applies to the conditional expectations as a whole rather than their individual parameters.

We want to recover the causal relations between $\widetilde{X}$ by removing batch $B$ and depth $N$ effects from the dataset because they correspond to the sequencing process rather than the underlying biology. We first consider removing sequencing depth by finding stably expressed housekeeping genes. Let $\widetilde{A}$ denote the set of housekeeping genes where $\widetilde{X}_i = \widetilde{x}_i$ is a constant for each $\widetilde{X}_i \in \widetilde{A}$; similarly $A$ refers to the corresponding set with Poisson measurement error. Let $N = n$ be large enough such that $\sum_{X_i \in A} x_i > 0$ for each sample. Then dividing by $L \triangleq \sum_{X_i \in A} X_i$ controls for sequencing depth in the following sense:

$$\lim_{N \to \infty} \frac{X_i}{\sum_{X_i \in A} X_i} = \lim_{N \to \infty} \frac{X_i/N}{\sum_{X_i \in A} X_i/N} = \frac{P_{ij}}{\sum_{X_i \in A} P_{ij}}$$

$$= \frac{\widetilde{X}_i \pi_{ij} / \sum_{i=1}^p \widetilde{X}_i \pi_{ij}}{\sum_{\widetilde{X}_i \in \widetilde{A}} \widetilde{x}_i \pi_{ij} / \sum_{i=1}^p \widetilde{X}_i \pi_{ij}} = \frac{\widetilde{X}_i \pi_{ij}}{\sum_{\widetilde{X}_i \in \widetilde{A}} \widetilde{x}_i \pi_{ij}},$$

where we have divided $\widetilde{X}_i \pi_{ij}$ by a constant in the last term. Thus, dividing by $L$ removes measurement error within each batch as $N \to \infty$. We assume that $N$ is so large that the approximation error is negligible. We only invoke the assumption in bulk RNA-seq, where the library size $N$ is on the order of at least tens of millions.

We do not divide by $L$ in practice because we may have $L = 0$ with finite $N$. We instead always include $L \cup B$ in the predictor set of downstream regressions. Conditioning on $L \cup B$ ensures that all downstream regressions mitigate depth and batch effects with adequate sequencing depth, or that $\mathbb{E}(Y|\widetilde{U}, B) = \mathbb{E}(Y|U, L, B)$ for any $\widetilde{U} \subseteq \widetilde{X}$ as $N \to \infty$. The equality holds almost surely under a mild smoothness condition:

Lemma 1. Assume Lipschitz continuity of the conditional expectation for all $N \geq n_0$:

$$\mathbb{E}\left|\mathbb{E}(Y|\widetilde{U}) - \mathbb{E}(Y|U, L, B)\right| \leq \mathbb{E}C_N\left|\widetilde{U} - \frac{U}{dL}\right|,$$

where $d = \frac{\pi_{UB}}{\sum_{\widetilde{X}_i \in \widetilde{A}} \widetilde{x}_i \pi_{iB}}$, $C_N \in O(1)$ is a positive constant, and we have taken an outer expectation on both sides. Then $\mathbb{E}(Y|\widetilde{U}) = \lim_{N\to\infty} \mathbb{E}(Y|U, L, B)$ almost surely.

We delegate proofs to the Appendix unless proven here in the Methods. Note that $\lim_{N\to\infty} \frac{U}{dL} = \widetilde{U}$, so the Lipschitz assumption intuitively means that accurate estimation of $\widetilde{U}$ implies accurate estimation of $\mathbb{E}(Y|\widetilde{U})$. Furthermore, conditioning on the library size $N$ instead of $L$ can introduce spurious dependencies because $N$ depends on all of the genes rather than just the stably expressed ones.

We now eliminate the need to condition on $L$. Note that $L$ is a sum of independent Poisson distributions given $B$ per Expression (4). This implies $Y \perp\!\!\!\perp L|(U, B)$ for any $N$, so that $\mathbb{E}(Y|\widetilde{U}) = \lim_{N\to\infty} \mathbb{E}(Y|U, L, B) = \lim_{N\to\infty} \mathbb{E}(Y|U, B)$ almost surely. We have proved:

Theorem 1. Consider the same assumption as Lemma 1. Then $\mathbb{E}(Y|\widetilde{U}) = \lim_{N\to\infty} \mathbb{E}(Y|U, B)$ almost surely, where we have eliminated the conditioning on $L$.

We emphasize again that these equalities hold for the conditional expectation but *not* for the regression parameters; the regression parameters do not converge in general unless we divide by $L$. We will only need to estimate conditional expectations in order to identify root causal genes.

## Identifying root causal genes

We showed how to overcome Poisson measurement error without sequencing depth normalization in the previous section. We leverage this technique to define a measure for identifying the root causal genes of $Y$.

## Definitions

A *root cause* of $Y$ corresponds to a root vertex that is an ancestor of $Y$ in $\mathbb{G}$. All root vertices are error terms in an augmented graph. We define the *root causal effect* of any $E_i \in E$ on $Y$ as $\Upsilon_i \triangleq \mathbb{P}(Y|E_i) - \mathbb{P}(Y)$ (**Strobl, 2024a**; **Strobl and Lasko, 2023c**).

We can identify root causes using the following result:

Proposition 1. If $E_i \not\perp\!\!\!\perp Y$ or $E_i \not\perp\!\!\!\perp Y|\text{Pa}(\widetilde{X}_i)$ (or both), then $E_i$ is a root cause of $Y$.

We can also claim the backward direction under d-separation faithfulness. We however avoid making this additional assumption because real biological data may not arise from distributions obeying d-separation faithfulness in practice (**Strobl, 2022a**).

Proposition 1 implies that $E_i$ is a root cause of $Y$ when:

$$\Delta_i \triangleq \mathbb{P}(Y|\text{Pa}(\widetilde{X}_i), E_i) - \mathbb{P}(Y|\text{Pa}(\widetilde{X}_i)) \neq 0.$$

The above quantity corresponds to the *conditional root causal effect* but not the root causal effect $\Upsilon_i$ due to the extra conditioning on $\text{Pa}(\widetilde{X}_i)$. The two terms may also differ in direction; if $\Delta_i > 0$, then this does not imply that $\Upsilon_i > 0$, and similarly for negative values. The two variables thus represent different quantities but – in terms of priority – we would estimate $\Upsilon_i$ when we have nonzero $\Delta_i$. Experimental results indicate that $\Upsilon_i$ and $\Delta_i$ take on similar values and agree in direction about 95% of the time in practice (Appendix).

We now encounter two challenges. First, the quantities $\Upsilon_i$ and $\Delta_i$ depend on the unknown error term $E_i$. We can however substitute $E_i$ with $\bar{X}_i$ in $\Delta_i$ due to the following result:

Proposition 2. We have $\mathbb{P}(Y|E_i, \text{Pa}(\widetilde{X}_i)) = \mathbb{P}(Y|\widetilde{X}_i, \text{Pa}(\widetilde{X}_i))$ under **Equation 3**.

We can thus compute the conditional root causal effect $\Delta_i$ without access to the error terms:

$$\Delta_i = \mathbb{P}(Y|\text{Pa}(\widetilde{X}_i), E_i) - \mathbb{P}(Y|\text{Pa}(\widetilde{X}_i))$$

$$= \mathbb{P}(Y|\text{Pa}(\widetilde{X}_i), \widetilde{X}_i) - \mathbb{P}(Y|\text{Pa}(\widetilde{X}_i)).$$

We can determine the root causal status of on when per Proposition 1. Nevertheless, the term 'root cause' in colloquial language refers to two concepts simultaneously: a root vertex that causes *and* has a large causal effect on . We thus say that is a *root causal gene* of if .

The second challenge involves computing the non-parametric probability distributions of which come at a high cost. We thus define the analogous expected version by:

$$\Gamma_i \triangleq \int y \left[ p(y \mid \mathrm{Pa}(\tilde{X}_i), \tilde{X}_i) - p(y \mid \mathrm{Pa}(\tilde{X}_i)) \right] dy$$

$$= \mathbb{E}(Y \mid \mathrm{Pa}(\tilde{X}_i), \tilde{X}_i) - \mathbb{E}(Y \mid \mathrm{Pa}(\tilde{X}_i))$$

$$= \mathbb{E}(Y \mid \mathrm{SP}(\tilde{X}_i), X_i, B) - \mathbb{E}(Y \mid \mathrm{SP}(\tilde{X}_i), B),$$

$$\Delta_i = \mathbb{P}(Y \mid \mathrm{Pa}(\tilde{X}_i), E_i) - \mathbb{P}(Y \mid \mathrm{Pa}(\tilde{X}_i))$$

$$= \mathbb{P}(Y \mid \mathrm{Pa}(\tilde{X}_i), \tilde{X}_i) - \mathbb{P}(Y \mid \mathrm{Pa}(\tilde{X}_i)).$$

where $p(Y)$ denotes the density of $Y$. Observe that if $\Delta_i = 0$, then $\Gamma_i = 0$. The converse is not true but likely to hold in real data when a change in the probability distribution also changes its expectation. The set $\mathrm{SP}(\tilde{X}_i) \subseteq X$ denotes the *surrogate parents* of $\tilde{X}_i$ corresponding to the variables in $X$ associated with $\mathrm{Pa}(\tilde{X}_i) \subseteq \tilde{X}$. The last equality holds almost surely as $N \to \infty$ by Theorem 1.

We call $\Phi_i \triangleq |\Gamma_i|$ the *Root Causal Strength* (RCS) of $\tilde{X}_i$ on $Y$. The RCS obtains a unique value $\Phi_i = \phi_{ij}$ for each patient $j$. We say that $\tilde{X}_i$ is a root causal gene of $Y$ for patient $j$ if $\phi_{ij} \gg 0$, since we posit a right skewed distribution of conditional root causal effects for each patient as in *Figure 1(b)*. We combine the RCS scores across a set of $n$ samples using the Deviation of the RCS (D-RCS) $\sqrt{\dfrac{1}{n} \sum_{j=1}^{n} \phi_{ij}^2}$,

or the deviation of RCS from zero. We may compute D-RCS for each cluster or globally across all patients depending on the context. We thus likewise say that $\tilde{X}_i$ is a root causal gene for a cluster of patients or all patients in a sample if its corresponding D-RCS score for the cluster or the sample is much lager than zero, respectively. Note that we do not specify a particular cutoff value for large (conditional) root causal effects, since the root causal effects likely lie on a continuous graduated scale as opposed to approximately two binary values. Nevertheless, visual inspection of the RCS or D-RCS histograms in disease should approximate a power law, where a large mass is concentrated around zero and a long tail extends to the right similar to folding *Figure 1b*.

## Algorithm

We now design an algorithm called Root Causal Strength using Perturbations (RCSP) that recovers the RCS scores using Perturb-seq and bulk RNA-seq data.

### Finding surrogate ancestors

Computing $\Phi_i$ for each $\tilde{X}_i \in \tilde{X}$ requires access to the surrogate parents of each variable or, equivalently, the causal graph $\mathbb{G}$. However, inferring $\mathbb{G}$ using causal discovery algorithms may lead to large statistical errors in the high dimensional setting (*Colombo, 2014*) and require restrictive assumptions such as d-separation faithfulness (*Spirtes et al., 2000*) or specific functional relations (*Peters, 2014*).

We instead directly utilize the interventional Perturb-seq data to recover a superset of the surrogate parents. We first leverage the global Markov property and equivalently write:

$$\Phi_i = \left| \mathbb{E}(Y \mid \mathrm{SA}(\tilde{X}_i), X_i, B) - \mathbb{E}(Y \mid \mathrm{SA}(\tilde{X}_i), B) \right|, \tag{5}$$

where $\mathrm{SA}(\tilde{X}_i)$ denotes the *surrogate ancestors* of $\tilde{X}_i$, or the variables in $X$ associated with the ancestors of $\tilde{X}_i$.

We discover the surrogate ancestors using unconditional independence tests. For any $X_k \in X$, we test $X_k \perp\!\!\!\perp P_i$ by unpaired two-sided t-test, where $P_i$ is an indicator function equal to one when we perturb $X_i$ and zero in the control samples of Perturb-seq. $P_i$ is thus a parent of $X_i$ alone but not a child of $B$, so we do not need to condition on $B$. We use the two-sided t-test to assess for independence because the t-statistic averages over cells to mimic bulk RNA-seq. If we reject the null and conclude that $X_k \not\perp\!\!\!\perp P_i$ then $X_k$ must be a descendant of $P_i$ by the global Markov property, so we include $X_k$ into the set of surrogate descendants $\mathrm{SD}(\tilde{X}_i)$. Curating every $X_j \in X$ such that $X_i \in \mathrm{SD}(\tilde{X}_j)$ into $\mathrm{SA}(\tilde{X}_i)$ yields the surrogate ancestors of $\tilde{X}_i$ as desired.

### Procedure

We now introduce an algorithm called Root Causal Strength using Perturbations (RCSP) that discovers the surrogate ancestors of each variable $\tilde{X}$ using Perturb-seq and then computes the RCS of each variable using bulk RNA-seq. We summarize RCSP in Algorithm 1.

RCSP takes Perturb-seq and bulk RNA-seq datasets as input. The algorithm first finds the surrogate descendants of each variable in $\widetilde{X}$ in Line 2 in order to identify the surrogate ancestors of each variable in Line 5. Access to the surrogate ancestors and the batches $B$ allows RCSP to compute $\Phi_i$ for each $X_i \in X$ from the bulk RNA-seq in Line 6. The algorithm thus outputs the RCS scores $\Phi$ as desired.

We certify RCSP as follows:

Theorem 2. (Fisher consistency) Consider the same assumption as Lemma 1. If unconditional d-separation faithfulness holds then RCSP recovers $\Phi$ almost surely as $N \rightarrow \infty$.

---

**Algorithm 1.** Root Causal Strength using Perturbations (RCSP).

---

**Input:** bulk RNA-seq data with batches $B$, Perturb-seq data
**Output:** RCS scores $\Phi$
1: **for** each $X_i \in X$ **do**
2:     $\mathrm{SD}(\widetilde{X}_i) \leftarrow$ all $X_k \in X$ s.t. $X_k \not\perp\!\!\!\perp P_i$ in Perturb-seq
3: **end for**
4: **for** each $X_i \in X$ **do**
5:     $\mathrm{SA}(\widetilde{X}_i) \leftarrow$ all $X_k \in X$ s.t. $X_i \in \mathrm{SD}(\widetilde{X}_k)$
6:     Compute $\Phi_i$ using **Equation 5** in bulk RNA-seq
7: **end for**

---

We engineered RCSP to only require *unconditional* d-separation faithfulness because real distributions may not obey full d-separation faithfulness (**Strobl, 2022a**).

## Synthetic data

### Simulations

We generated a linear SEM obeying **Equation 3** specifically as $\widetilde{X}_i = \widetilde{X}\beta_i + E_i$ for every $\widetilde{X}_i \in \widetilde{X}$ and similarly $Y = \widetilde{X}\beta_Y + E_Y$. We included $p + 1 = 2500$ variables in $\widetilde{X} \cup Y$. We instantiated the coefficient matrix $\beta$ by sampling from a $\mathrm{Bernoulli}(2/(p-1))$ distribution in the upper triangular portion of the matrix. The resultant causal graph thus has an expected neighborhood size of 2. We then randomly permuted the ordering of the variables. We introduced weights into the coefficient matrix by multiplying each entry in $\beta$ by a weight sampled uniformly from $[-1, -0.25] \cup [0.25, 1]$. The error terms each follow a standard Gaussian distribution multiplied by 0.5. We introduced batch effects by drawing each entry of the mapping efficiencies $\pi$ from the uniform distribution between 10 and 1000 for the bulk RNA-seq, and between 0.1 and 1 for the Perturb-seq. We set $\widetilde{X}_i \leftarrow \mathrm{softplus}(\widetilde{X}_i)$ and then obtained the corrupted surrogate $X_i$ distributed $\mathrm{Pois}(\widetilde{X}_i\pi_{ij})$ for each $\widetilde{X}_i \in \widetilde{X}$ and batch $j$. We chose $Y$ uniformly at random from the set of vertices with at least one parent and no children. We drew 200 samples for the bulk RNA-seq data to mimic a large but common dataset size. We introduced knockdown perturbations in Perturb-seq by subtracting an offset of two in the softplus function: $\widetilde{X}_i \leftarrow \mathrm{softplus}(\widetilde{X}_i - 2)$. We finally drew 200 samples for the control and each perturbation condition to generate the Perturb-seq data. We repeated the above procedure 30 times.

### Comparators

We compared RCSP against the following four algorithms:

1. Additive noise model (ANM) (**Peters, 2014**; **Strobl and Lasko, 2023a**): performs non-linear regression of $X_i$ on $\mathrm{Pa}(X_i) \cup B$ and then regresses $Y$ on the residuals $E \setminus E_i$ to estimate $\left| \mathbb{E}(Y|E \setminus E_i) - \mathbb{E}(Y|X, B) \right|$ for each $X_i \in X$. The non-linear regression residuals are equivalent to the error terms assuming an additive noise model.
2. Linear Non-Gaussian Acyclic Model (LiNGAM) (**Peters, 2014**; **Strobl and Lasko, 2022b**): same as above but performs linear instead of non-linear regression.
3. CausalCell (**Wen et al., 2023**): selects the top 50 genes with maximal statistical dependence to $Y$, and then runs the Peter-Clark (PC) algorithm (**Spirtes et al., 2000**) using a non-parametric conditional independence test to identify a causal graph among the top 50 genes. The algorithm does not perform root causal inference, so we use ANM as above but condition on the estimated parent sets for the top 50 genes and the ancestors inferred from the Perturb-seq data otherwise
4. Univariate regression residuals (Uni Reg): regresses $Y$ on $X_i \cup B$ and estimates the absolute residuals $\left| Y - \mathbb{E}(Y|X_i, B) \right|$ for each $X_i \in X$.
5. Multivariate regression residuals (Multi Reg): similar to above but instead computes the absolute residuals after regressing $Y$ on $(X \setminus X_i) \cup B$.

The first two methods are state-of-the-art approaches used for root causal discovery. Univariate and multivariate regressions do not distinguish between predictivity and causality, but we included them as sanity checks. We performed all non-linear regressions using multivariate adaptive regression splines to control for the underlying regressor (*Friedman, 1991*). We also standardized all variables before running the regressions to prevent gaming of the marginal variances in causal discovery (*Reisach, 2021*; *Ng, 2024*). We compared the algorithms on their accuracy in estimating $\Phi$.

## Real data

### Quality control

We downloaded Perturb-seq datasets of retinal pigment epithelial cells from the RPE-1 cell line, and myeloid leukemia cells from the K562 cell line (*Replogle et al., 2022*). We used the genome-wide dataset version for the latter. We downloaded the datasets from the scPerturb database on Zenodo (*Green, 2022*) with the same quality controls as the original paper. Replogle et al. computed adjusted library sizes by equalizing the mean library size of control cells within each batch. Cells with greater than a 2000 or 3000 library size, and less than 25% or 11% mitochondrial RNA were kept, respectively. The parameters were chosen by plotting the adjusted library sizes against the mitochondrial RNA counts and then manually setting thresholds that removed low-quality cells likely consisting of ambient mRNA transcripts arising from premature cell lysis or cell death.

We next downloaded bulk RNA-seq datasets derived from patients with age-related macular degeneration (AMD; GSE115828) and multiple sclerosis (MS; GSE137143) (*Ratnapriya et al., 2019*; *Kim et al., 2021*). We excluded 10 individuals from the AMD dataset including one with an RNA integrity number of 21.92, five missing an integrity number (all others had an integrity number of less than 10), and four without a Minnesota Grading System score. We kept all samples from the MS dataset derived from CD4+ T cells but filtered out genes with a mean of less than 5 counts as done in the original paper.

We finally kept genes that were present in both the AMD bulk dataset and the RPE-1 Perturb-seq dataset, yielding a final count of 513 bulk RNA-seq samples and 247,914 Perturb-seq samples across 2077 genes. We also kept genes that were present in both the MS bulk dataset and the K562 Perturb-seq dataset, yielding a final count of 137 bulk RNA-seq samples and 1,989,578 Perturb-seq samples across 6882 genes. We included age and sex as a biological variable as covariates for every patient in both datasets in subsequent analyses.

### Evaluation rationale

We do not have access to the ground truth values of $\Phi$ in real data. We instead evaluate the RCSP estimates of $\Phi$ using alternative sources of ground truth knowledge. We first assess the accuracy of RCS using the control variable age as follows:

1. Determine if the RCS values of age identify age as a root cause with large causal effect in diseases that progress over time.Second, few root causal genes should drive pathogenesis because the effects of a few error terms distribute over many downstream genes. We verify the sparsity of root causal genes as follows:
2. Determine if the distribution of D-RCS concentrates around zero more than the distribution of the Deviation of Statistical Dependence (D-SD) defined as $\sqrt{\frac{1}{n}\sum_{j=1}^{n}\omega_{ij}^2}$ for each gene $\widetilde{X}_i \in \widetilde{X}$ where $\Omega_i = \left| \mathbb{E}(Y|X_i, B) - \mathbb{E}(Y|B) \right|$ and $\omega_{ij}$ its value for patient $j$. Determine if genes with the top D-RCS scores correspond to genes known to cause the disease. Despite the sparsity of root causal genes, we still expect the root causal genes to correspond to at least some known causes of disease:
3. Determine if genes with the top D-RCS scores correspond to genes known to cause the disease. Next, the root causal genes initiate the vast majority of pathogenesis, and we often have knowledge of pathogenic pathways even though we may not know the exact gene expression cascade leading to disease. Intervening on root causal genes should also modulate patient symptoms. We thus further evaluate the accuracy of RCSP using pathway and drug enrichment analyses as follows:
4. Determine if the D-RCS scores identify known pathogenic pathways of disease in pathway enrichment analysis.
5. Determine if the D-RCS scores identify drugs that treat the disease. Finally, complex diseases frequently involve multiple pathogenic pathways that differ between

patients. Patients with the same complex disease also respond differently to treatment. We hence evaluate the precision of RCS as follows:

6. Determine if the patient-specific RCS scores identify subgroups of patients involving different but still known pathogenic pathways.
7. Determine if the patient-specific RCS scores identify subgroups of patients that respond differently to drug treatment.

In summary, we evaluate RCSP in real data based on its ability to (1) identify age as a known root cause, (2) suggest an omnigenic root causal model, (3) recover known causal genes, (4) find known pathogenic pathways, (5) find drugs that treat the disease, and (6,7) delineate patient subgroups.

### Enrichment analyses

Multivariate adaptive regression splines introduce sparsity, but enrichment analysis performs better with a dense input. We can estimate the conditional expectations of $\Phi$ using any general non-linear regression method, so we instead estimated the expectations using kernel ridge regression equipped with a radial basis function kernel (*Shawe-Taylor and Cristianini, 2004*). We then computed the D-RCS across all patients for each variable in $X$. We ran pathway enrichment analysis using the fast gene set enrichment analysis (FGSEA) algorithm (*Sergushichev, 2016*) with one hundred thousand simple permutations using the D-RCS scores and pathway information from the Reactome database (version 1.86.0; *Fabregat et al., 2017*). We likewise performed drug set enrichment analysis with the Drug Signature database (version 1.0; *Yoo et al., 2015*). We repeated the above procedures for the D-RCS of all clusters identified by hierarchical clustering via Ward's method (*Ward, 1963*).

## Acknowledgements

Research reported in this report was supported by the National Human Genome Research Institute of the National Institutes of Health under award numbers R01HG011138 and R35HG010718.

## Additional information

### Funding

| Funder | Grant reference number | Author |
| --- | --- | --- |
| National Human Genome Research Institute | R01HG011138 | Eric Gamazon |
| National Human Genome Research Institute | R35HG010718 | Eric Gamazon |

The funders had no role in study design, data collection and interpretation, or the decision to submit the work for publication.

### Author contributions

Eric V Strobl, Conceptualization, Resources, Data curation, Software, Formal analysis, Validation, Investigation, Visualization, Methodology, Writing - original draft, Writing - review and editing; Eric Gamazon, Supervision, Writing - review and editing

### Author ORCIDs

Eric V Strobl  https://orcid.org/0009-0003-9894-9694

Reviewer #1 (Public review): https://doi.org/10.7554/eLife.100949.3.sa1
Reviewer #2 (Public review): https://doi.org/10.7554/eLife.100949.3.sa2
Author response https://doi.org/10.7554/eLife.100949.3.sa3

# Additional files

**Supplementary files**
MDAR checklist

**Data availability**
All datasets analyzed in this study have been previously published and are publicly accessible as follows: Bulk RNA-seq for AMD: GSE1158282. Bulk RNA-seq for MS: GSE1371433. Perturb-seq for the RPE-1 and K562 cell lines: DOI 10044268. R code needed to replicate all experimental results is available on GitHub (copy–archived at *Strobl, 2024b*).

The following previously published datasets were used:

| Author(s) | Year | Dataset title | Dataset URL | Database and Identifier |
|---|---|---|---|---|
| Ratnapriya R, Starostik M, Kayode S, Kwicklis M, Kapphahn R, Fritsche L, Walton A, Arvanitis M, Geiser L, Pietraszkiewicz A, Montezuma S, Chew E, Battle A, Abecasis G, Ferrington D, Chatterjee N, Swaroop A | 2019 | Integrated analysis of genetic variants regulating retinal transcriptome (GREx) identifies genes underlying age-related macular degeneration | https://www.ncbi.nlm.nih.gov/geo/query/acc.cgi?acc=GSE115828 | NCBI Gene Expression Omnibus, GSE115828 |
| Kim K, Baranzini SE | 2020 | Cell-Type-Specific Transcriptome of CD4+, CD8+ T cells and CD14+ monocytes in multiple sclerosis | https://www.ncbi.nlm.nih.gov/geo/query/acc.cgi?acc=GSE137143 | NCBI Gene Expression Omnibus, GSE137143 |
| Replogle JM, Saunders RA, Pogson AN, Hussmann JA, Lenail A, Guna A, Mascibroda L, Wagner EJ, Adelman K, Lithwick-Yanai G, Iremadze N, Oberstrass F, Lipson D, Bonnar JL, Jost M, Norman TM, Weissman JS | 2022 | Mapping information-rich genotype-phenotype landscapes with genome-scale Perturb-seq | https://doi.org/10.5281/zenodo.10044268 | Zenodo, 10.5281/zenodo.10044268 |

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

# Appendix 1

## Additional synthetic data results

### Normalization by sequencing depth

We theoretically showed that RCS does not require normalization by sequencing depth in the Methods using an asymptotic argument. We tested this claim empirically by drawing 200 bulk RNA-seq samples from random DAGs as in the Methods but over $p + 1 = 250$ variables. We varied the mean sequencing depth $N/p$ of each gene from 15, 20, 30, 50, 90, 170, 330–650 counts; multiplying $N/p$ by $p$ recovers the library size $N$. We only included one batch in the bulk RNA-seq in order to isolate the effect of sequencing depth. We compared no normalization, normalization by 10 housekeeping genes, normalization by 20 housekeeping genes, and normalization by library size. We repeated each experiment 100 times and thus generated a total of 100×4×8=3200 datasets.

We plot the results in *Appendix 1—figure 1*. All methods improved with increasing mean sequencing depth as expected. The no normalization strategy performed the best at low mean sequencing depths, followed by the housekeeping genes and then total library size. The result even held with a small library size of $N = 15 \times 249 = 3735$ at the smallest mean sequencing depth of 15, suggesting that the asymptotic argument holds well in bulk RNA-seq where $N/p$ is often greater than 500 and $N$ greater than the tens of millions. However, the average RMSEs of all normalization methods became more similar as sequencing depth increased. We conclude that no normalization exceeds or matches the accuracy of other strategies. We therefore do not normalize by sequencing depth in subsequent analyses.

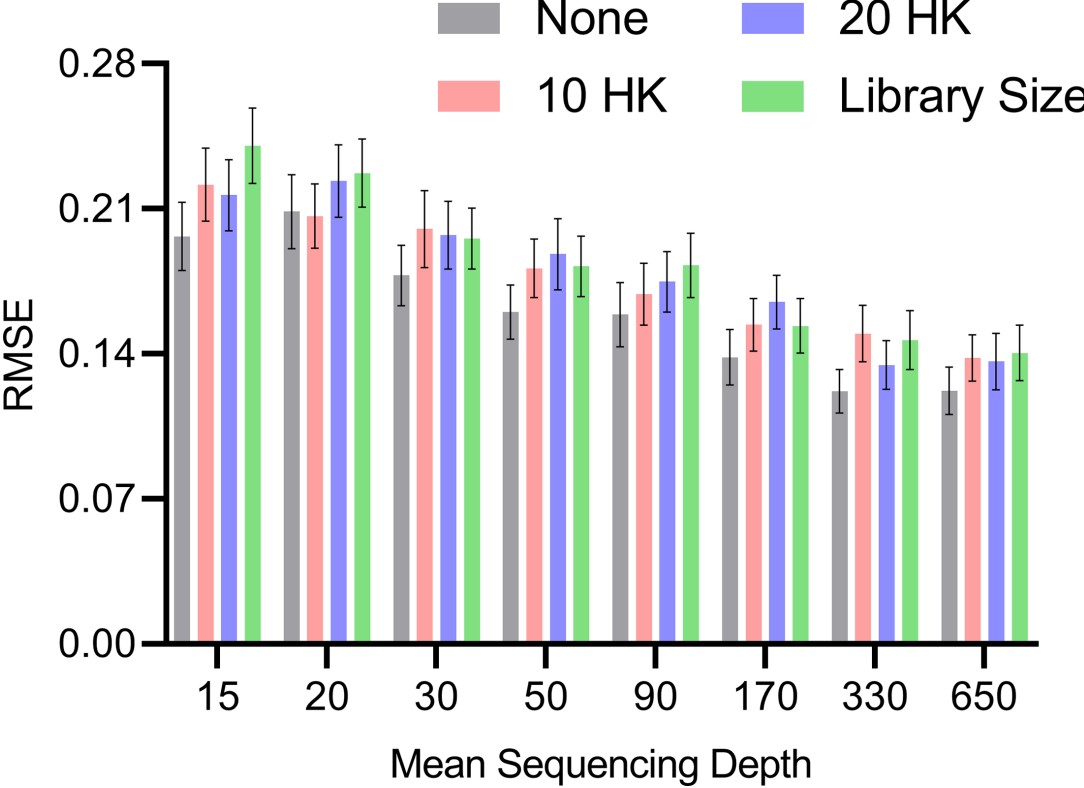

**Appendix 1—figure 1.** Mean RMSE to the ground truth RCS values across different mean sequencing depths and normalization strategies. The no normalization strategy achieved low RMSEs at lower mean sequencing depths, but the performances of all methods converged as the mean sequencing depths increased. Error bars denote 95% confidence intervals of the mean over 100 datasets.

### Functional causal models and measurement error

The experiments in the Results section quantify the accuracies of the algorithms in estimating Φ. However, the functional causal models ANM and LiNGAM also estimate the error terms as an

intermediate step, whereas RCSP does not. We therefore also investigated the accuracies of ANM and LiNGAM in estimating the error term values.

Theoretical results suggest that ANM and LiNGAM cannot consistently estimate the error terms in RNA-seq due to the Poisson measurement error. We empirically tested this hypothesis by sampling from bulk RNA-seq data as in the Methods but with $p + 1 = 100$ and a batch size of one in order to isolate the effect of measurement error. We repeated the experiment 100 times for bulk RNA-seq sample sizes of 100, 200, 400, 800, 1600 and 3200. We plot the results in *Appendix 1—figure 2*. The accuracies of ANM and LiNGAM did not improve beyond an RMSE of 0.44 to the ground truth error term values even with a large sample size of 6400. We conclude that ANM and LiNGAM cannot estimate the error terms accurately in the presence of measurement error even with large sample sizes.

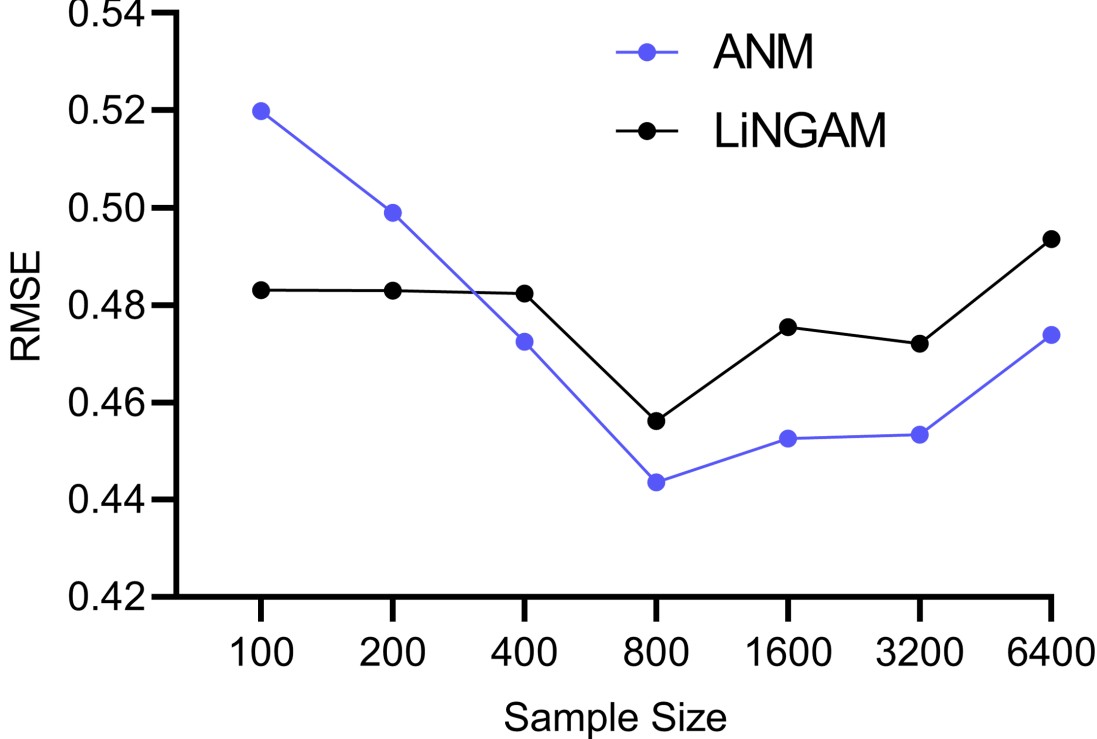

**Appendix 1—figure 2.** Mean RMSE values to the ground truth error term values across different sample sizes. The accuracies of ANM and LiNGAM do not improve with increasing sample sizes.

## Cyclic causal graphs

We also evaluated the algorithms on directed graphs with cycles. We generated a linear SEM over $p + 1 = 1000$ variables in $\widetilde{X} \cup Y$. We sampled the coefficient matrix $\beta$ from a Bernoulli $(1/(p-1))$ distribution but did not restrict the non-zero coefficients to the upper triangular portion of the matrix. We then proceeded to permute the variable ordering and weight each entry as in the Methods for the DAG. We repeated this procedure 30 times and report the results in *Appendix 1—figure 3*.

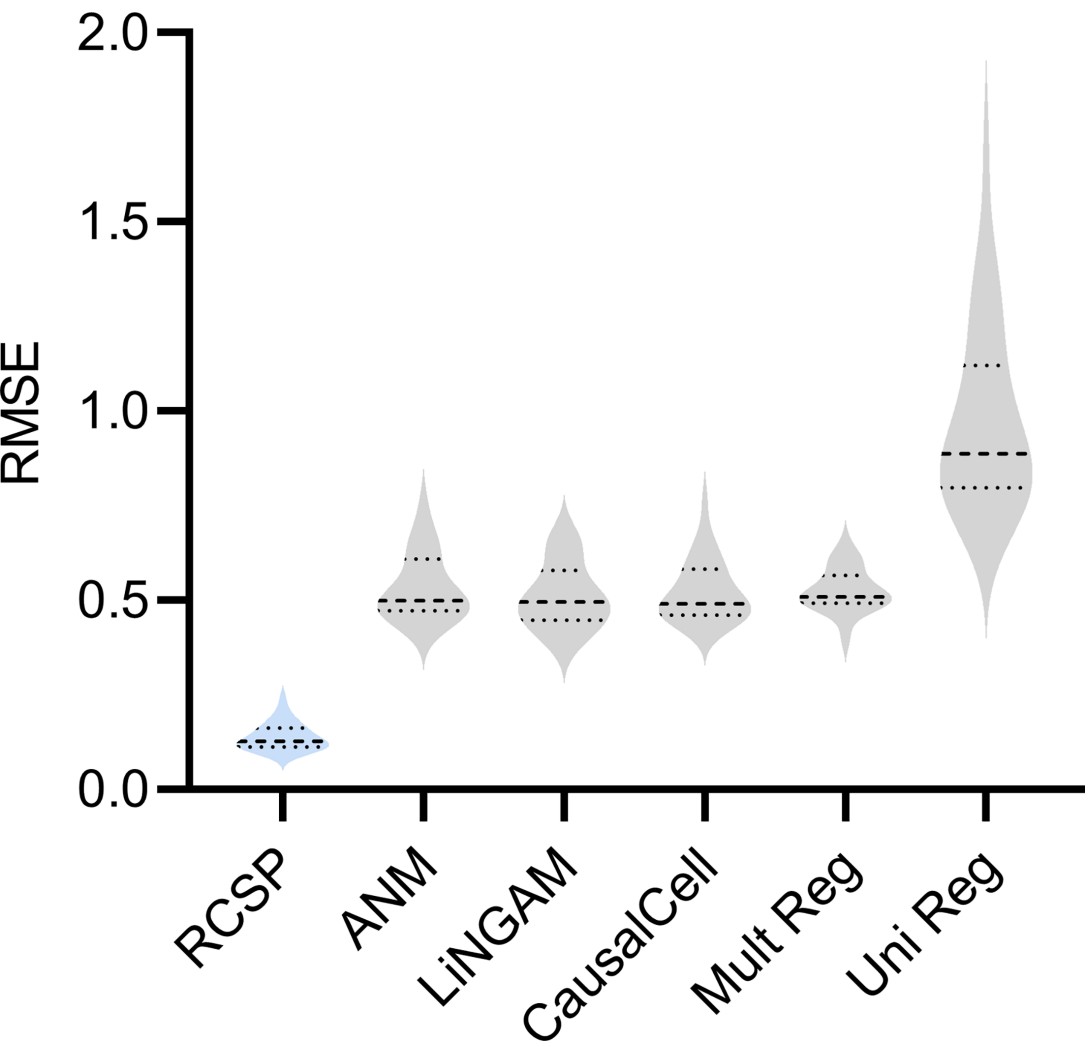

**Appendix 1—figure 3.** RCSP achieved the lowest RMSE in cyclic graphs as well. However, error terms can influence ancestors in the cyclic case, so the interpretation of the RCS remains unclear when cycles exist.

RCSP again outperformed all other algorithms even in the cyclic case. The results suggest that conditioning on the surrogate ancestors also estimates the RCS well even in the cyclic case. However, we caution that an error term $E_i$ can affect the ancestors of $\bar{X}_i$ when cycles exist. As a result, the RCS may not isolate the causal effect of the error term and thus not truly coincide with the notion of a root causal effect in cyclic causal graphs.

## DAG Incongruence

We next assessed the performance of RCSP when the DAG underlying the Perturb-seq data differs from the DAG underlying the bulk RNA-seq data. We considered a mixture of two random DAGs in bulk RNA-seq, where one of the DAGs coincided with the Perturb-seq DAG and the second alternate DAG did not. We instantiated and simulated samples from each DAG as per the previous subsection. We generated 0%, 25%, 50%, 75%, and 100% of the bulk RNA-seq samples from the alternate DAG, and the rest from the Perturb-seq DAG. We ideally would like to see the performance of RCSP degrade gracefully, as opposed to abruptly, as the percent of samples derived from the alternate DAG increases.

We summarize results in *Appendix 1—figure 4*. As expected, RCSP performed the best when we drew all samples from the same underlying DAG for Perturb-seq and bulk RNA-seq. However, the performance of RCSP also degraded slowly as the percent of samples increased from the alternate DAG. We conclude that RCSP can accommodate some differences between the underlying DAGs in Perturb-seq and bulk RNA-seq with only a mild degradation in performance.

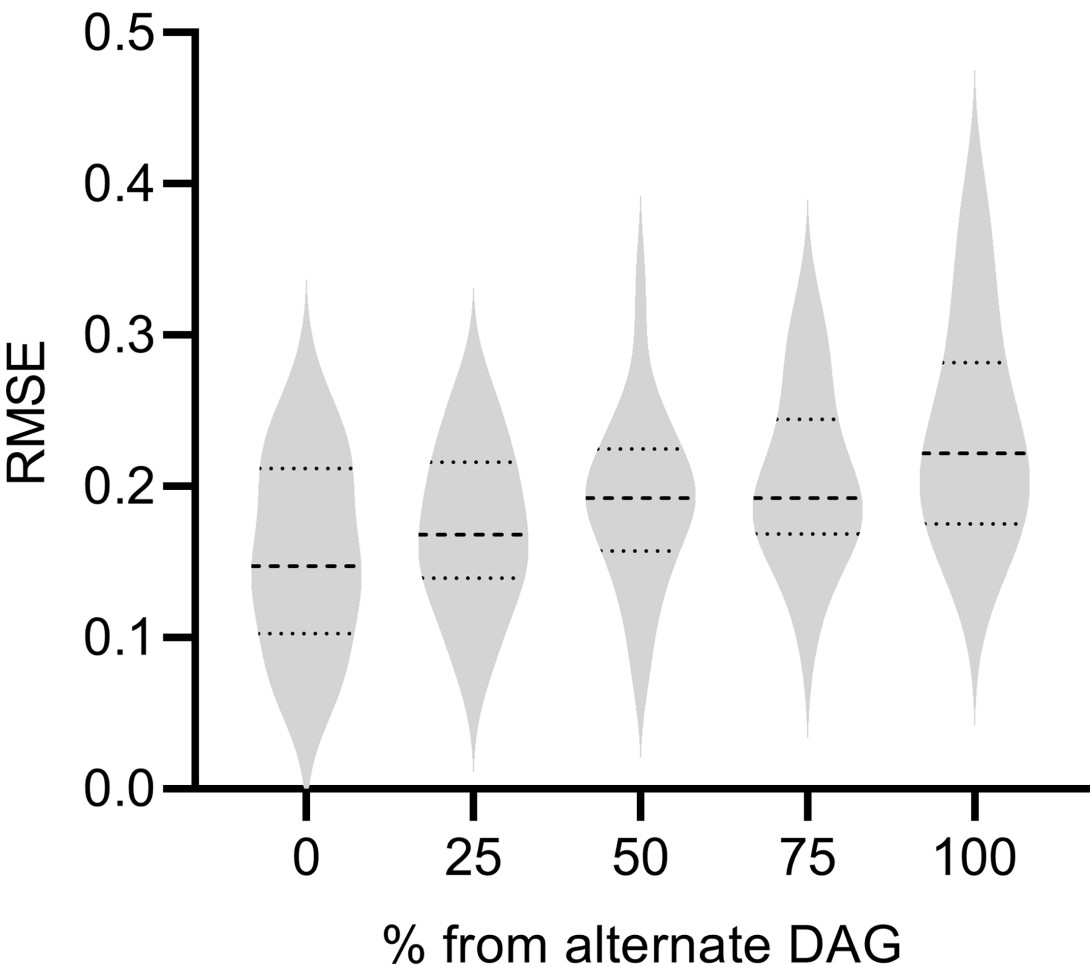

**Appendix 1—figure 4.** The performance of RCSP degrades gracefully as the percent of samples from the alternate DAG increases.

## Non-sink target

We considered the scenario where $Y$ is a non-sink (or non-terminal) vertex. If $Y$ is a parent of a gene expression level, then we cannot properly condition on the parents because modern Perturb-seq datasets usually do not intervene on $Y$ or measure $Y$. We therefore empirically investigated the degradation in performance resulting from a non-sink target $Y$, in particular for gene expression levels where $Y$ is a parent. We simulated 200 samples from bulk RNA-seq and each condition of Perturb-seq with a DAG over 1000 vertices, an expected neighborhood size of 2 and a non-sink target $Y$. We then removed the outgoing edges from $Y$ and resampled the DAG with a sink target. We compared the results of RCSP for both DAGs in gene expression levels where $Y$ is a parent before removing the outgoing edges from $Y$. We plot the results in *Appendix 1—figure 5*. As expected, we observe a degradation in performance when $Y$ is not a sink vertex, where the mean RMSE increased from 0.045 to 0.342. We conclude that RCSP is sensitive to violations of the sink target assumption.

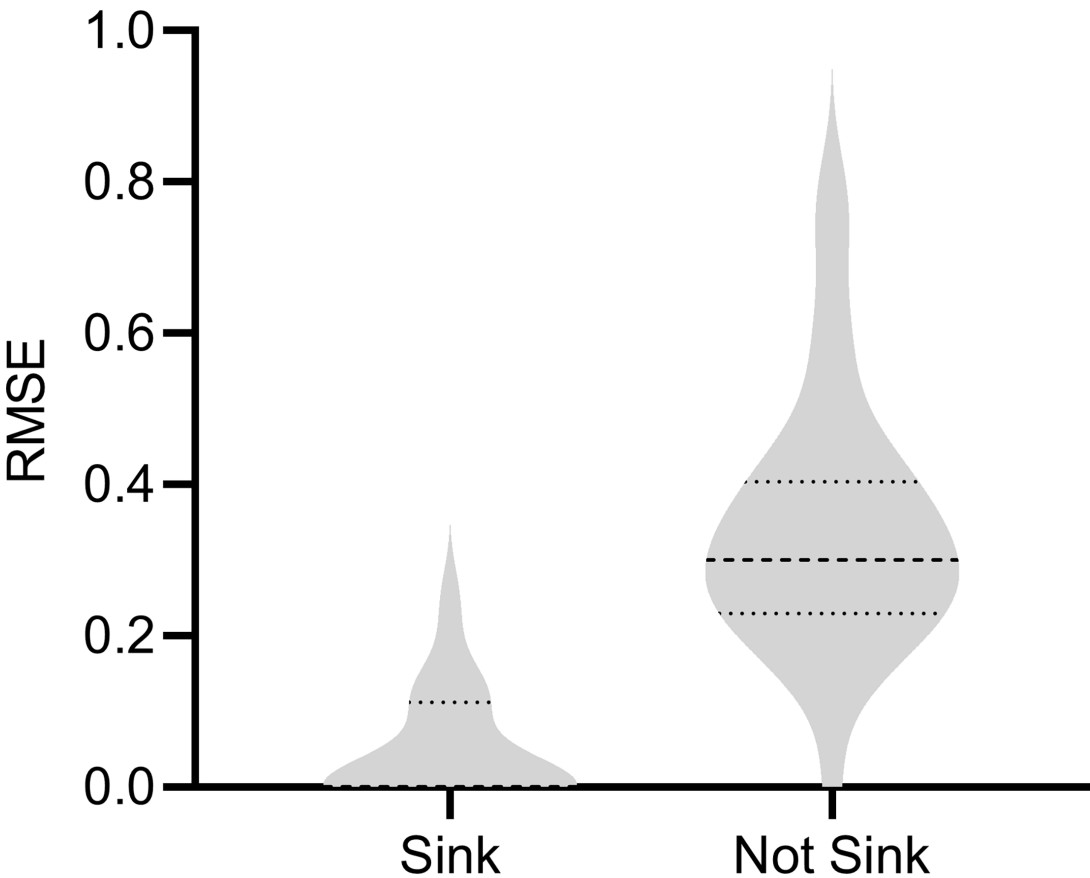

**Appendix 1—figure 5.** Results with a sink or non-sink target $Y$. RCSP estimated the RCS scores less accurately with a non-sink target indicating that the algorithm is sensitive to violations of the sink target assumption.

### Root causal effect versus conditional root causal effect

We compared the expected and unconditional root causal effects $\Omega_i \triangleq \mathbb{E}(Y|E_i) - \mathbb{E}(Y)$ to the expected and conditional root causal effects or, equivalently, the signed RCS scores $\Gamma$. These root causal effects and conditional root causal effects are not the same, but they are similar. We empirically investigated the differences between the estimated values of $\Gamma$ and the true values of $\Omega$ using the RMSE and also the percent of samples with incongruent signs; $\Gamma$ and $\Omega$ have incongruent signs if one is positive and the other is negative. We again drew 200 bulk RNA-seq samples from random DAGs as in the Methods over $p + 1 = 250$ variables with one batch. We varied the bulk RNA-seq sample size from 100, 200, 400–800. We also compared true $\Gamma$ against true $\Omega$ by estimating the two to negligible error using 20,000 samples of $\widetilde{X}$. We repeated each experiment 100 times and thus generated a total of $100 \times 5 = 500$ datasets.

We summarize the results in *Appendix 1—figure 6*. The estimated $\Gamma$ values approached the true $\Omega$ values with increasing sample sizes. The true $\Gamma$ values did not converge exactly to the true $\Omega$ values, but the RMSE remained low at 0.05 and the two values differed in sign only around 5.3% of the time. Increasing the number of samples of $\widetilde{X}$ to 50,000 did not change performance, confirming that we reached the floor. We conclude that the empirical results replicate the theoretical results because $\Gamma$ and $\Omega$ do not match exactly. However, the two quantities take on similar values and their signs matched around 95% of the time in practice.

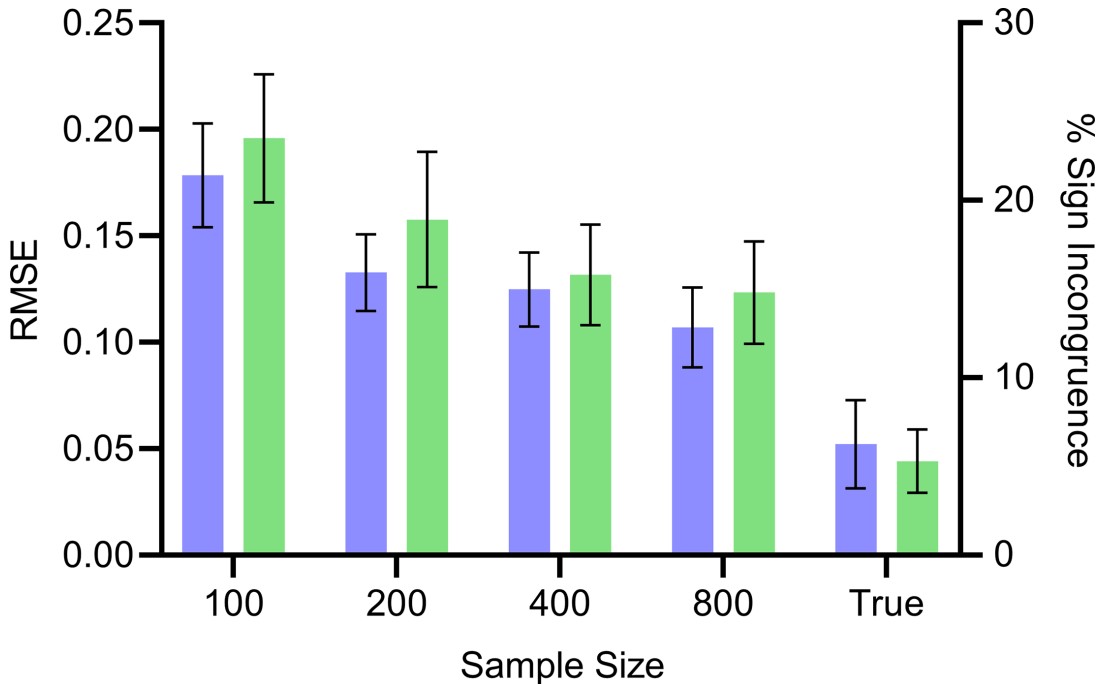

**Appendix 1—figure 6.** Mean RMSE (blue, left) and percent sign incongruence (green, right) of the expected root causal effects and signed RCS values, respectively. The RMSE continues to decrease with increasing sample size but reaches a floor of around 0.05. Similarly, the percent sign incongruence decreases but reaches a floor of around 5%. Error bars denote 95% confidence intervals of the mean over 100 datasets.

## Additional results for age-related macular degeneration

### Algorithm comparisons

We say that an algorithm performs well in real data if it simultaneously (1) identifies a sparse set of root causal genes, (2) recovers known pathogenic pathways with high specificity measured by the sparsity of leading edge genes, and (3) clusters patients into clear subgroups.

We compared the algorithms with the AMD data. We summarize the results in *Appendix 1—figure 7* plotted on the next page. The figure contains 6 rows and 3 columns. Similar to the D-RCS, we can compute the standard deviation of the output of each algorithm from zero for each gene. The first column in *Appendix 1—figure 7* denotes the histograms of these standard deviations across the genes. We standardized the outputs to have mean zero and unit variance. We then added the minimum value so that all histograms begin at zero; note that the bars at zero are not visible for many algorithms, since only a few genes attained standard deviations near the minimum. If an algorithm accurately identifies root causal genes, then it should only identify a few genes with large conditional root causal effects under the omnigenic root causal model. The RCSP algorithm had a histogram with large probability mass centered around zero with a long tail to the right. The standard deviations of the outputs of the other algorithms attained large values for nearly all genes. Incorporating feature selection and causal discovery with CausalCell introduced more outliers in the histogram of ANM. We conclude that only RCSP detected an omnigenic root causal model.

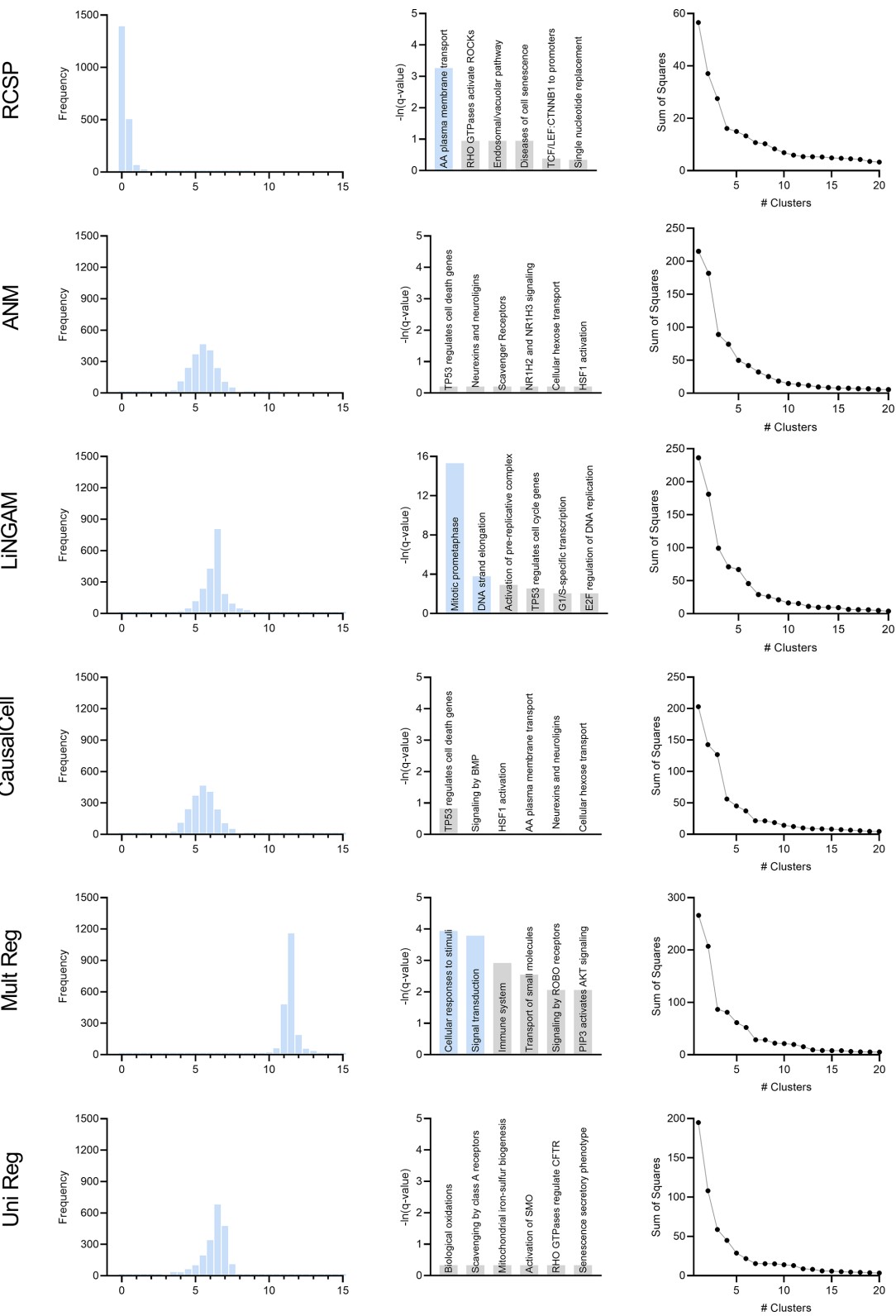

**Appendix 1—figure 7.** Comparison of the algorithms in age-related macular degeneration.

We plot the results of pathway enrichment analysis in the second column of *Appendix 1—figure 7*. RCSP, LiNGAM and univariate regression detected pathways related to oxidative stress in AMD. However, the 'mitotic prometaphase' and 'DNA strand elongation' pathways in blue for LiNGAM involved 94 and 27 leading edge genes, respectively. The 'cellular responses to stimuli' and 'signal

transduction' pathways for multivariate regression also involved 253 and 282 leading edge genes. In contrast, the 'amino acid plasma membrane transport' pathway for RCSP involved two leading edge genes. We conclude that RCSP identified a known pathogenic pathway of AMD with the fewest number of leading edge genes.

We finally plot the clustering results in the third column of *Appendix 1—figure 7*. The RCSP sum of squares plot revealed a sharp elbow at four groups of patients, whereas the other plots did not reveal a clear number of categories using the elbow method. We conclude that only RCSP identified clear subgroups of patients in AMD.

In summary, RCSP detected a small set of root causal genes, identified pathogenic pathways with maximal specificity and discovered distinguishable patient subgroups. We therefore conclude that RCSP outperformed all other algorithms in the AMD dataset.

## Effect of sequencing depth

Theorem 1 states that RCS scores may exhibit bias with insufficient sequencing depth. The genes with large D-RCS scores may therefore simply have low sequencing depths. To test this hypothesis, we plotted sequencing depth against D-RCS scores. Consistent with Theorem 1, we observed a small negative correlation between D-RCS and sequencing depth ($\rho$ = −0.16, *P*=2.04E-13), and D-RCS scores exhibited greater variability at the lowest sequencing depths (*Appendix 1—figure 8*). However, genes with the largest D-RCS scores had mean sequencing depths interspersed between 20 and 3000. We conclude that genes with the largest D-RCS scores had a variety of sequencing depths ranging from low to high.

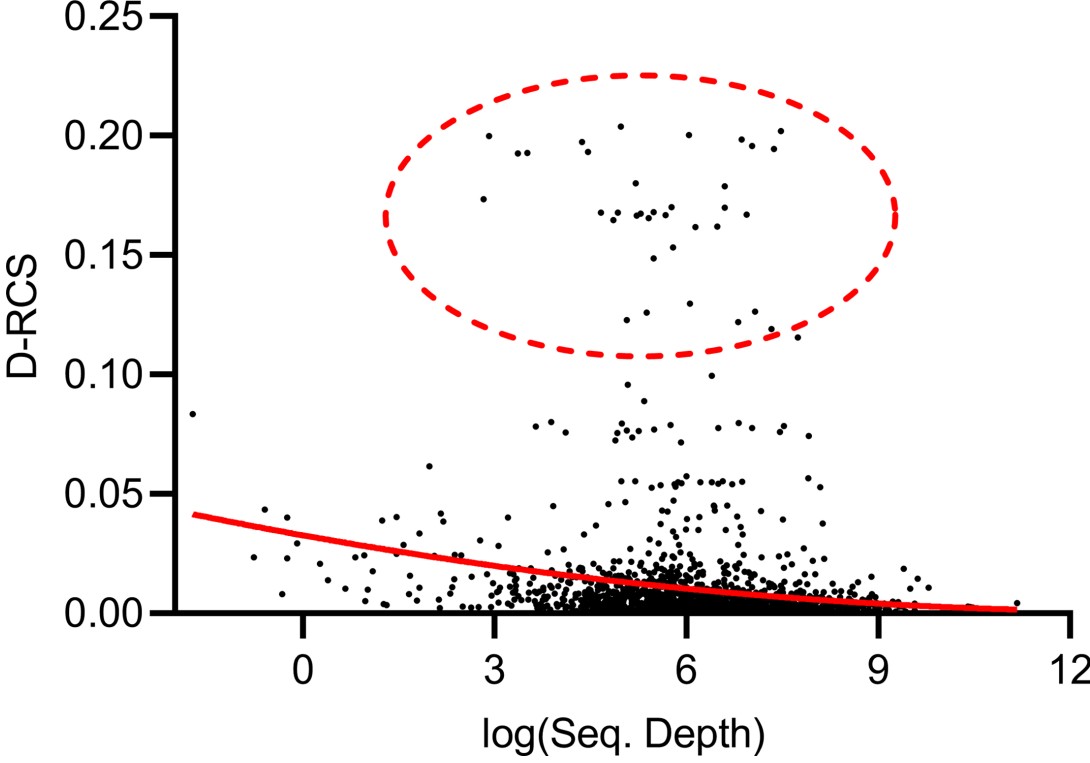

**Appendix 1—figure 8.** Mean sequencing depth of each gene plotted against their D-RCS scores in AMD. Genes with the largest D-RCS scores (red ellipse) had a variety of sequencing depths.

## Biological results

We provide the full pathway enrichment analysis results in *Appendix 1—table 1* corresponding to *Figure 3c*. We summarize pathway enrichment analysis of the black cluster of *Figure 3g* in *Figure 3j*. However, analyses of the blue, green and pink clusters did not yield significant pathways even at a liberal FDR threshold of 10%.

**Appendix 1—table 1.** Full pathway enrichment analysis results for all patients in the AMD dataset. We list the Entrez gene IDs of up to the top three leading edge genes in the right-most column.

| Pathway | p-value | q-value | Effect Size | Leading Edge |
|---|---|---|---|---|
| Amino acid transport across the plasma membrane | 2.44e-05 | 0.038 | 0.995 | 81,406,510 |
| RHO GTPases Activate ROCKs | 2.09e-03 | 0.388 | 0.976 | 46,595,500 |
| Endosomal/Vacuolar pathway | 2.32e-03 | 0.388 | 0.998 | 3107 |
| Diseases of Cellular Senescence | 2.97e-03 | 0.388 | 0.997 | 1021 |
| Binding of TCF/LEF:CTNNB1 to target gene promoters | 6.52e-03 | 0.68 | 0.993 | 4609 |
| APEX1-Indep. Resolution of AP Sites via Nucleotide Replacement | 7.28e-03 | 0.712 | 0.980 | 112,847,515 |
| MASTL Facilitates Mitotic Progression | 1.59e-02 | 0.978 | 0.911 | 84,930,983 |
| PI5P Regulates TP53 Acetylation | 1.94e-02 | 0.978 | 0.980 | 79837 |
| Formation of Incision Complex in GG-NER | 2.24e-02 | 0.978 | 0.791 | 296,699,782,967 |
| Glycine degradation | 2.24e-02 | 0.978 | 0.977 | 1738 |
| Prefoldin mediated transfer of substrate to CCT/TriC | 3.96e-02 | 0.978 | 0.787 | 5,203,520,110,576 |

We examined whether the clusters of *Figure 3g* differentiate dry and wet macular degeneration. Wet macular degeneration is associated with the highest Minnesota Grading System (MGS) score of 4 (*Olsen and Feng, 2004*). We plotted the UMAP embedding against MGS (*Appendix 1—figure 9a*). None of the two UMAP dimensions correlated significantly with the MGS score (5% uncorrected threshold by Spearman's correlation test). These results and the large RCS scores of age in *Figure 3a* seem to support the hypothesis that wet macular degeneration is a more severe type of dry macular degeneration. However, MGS does not differentiate between wet macular degeneration and late stage dry macular degeneration involving geographical atrophy. We therefore cannot separate late stage dry and wet macular degeneration using the RCS scores alone.

We correlated the two UMAP dimensions with the top 30 genes ranked by their RCS scores. We plot genes with the highest correlation to the first and second UMAP dimensions in *Appendix 1—figure 9b and c*, respectively. Many genes correlated with the first dimension, but only three genes correlated with the second at an FDR threshold of 5%.

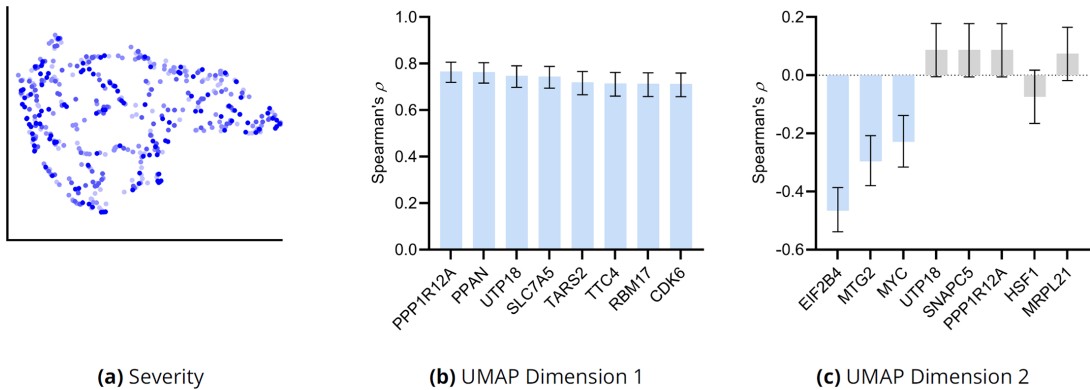

**(a)** Severity **(b)** UMAP Dimension 1 **(c)** UMAP Dimension 2

**Appendix 1—figure 9.** Additional UMAP embedding results for AMD. (**a**) The UMAP dimensions did not correlate with AMD severity as assessed by the MGS score. Many genes correlated with the first UMAP dimension in (**b**), but only three genes correlated with the second UMAP dimension in (**c**). Blue bars passed an FDR threshold of 5%, error bars denote 95% confidence intervals, and the sample size corresponded to 513 individuals.

We finally performed drug enrichment analysis in each of the four clusters in *Figure 3g*. We summarize the results in *Appendix 1—figure 10*. Only two drugs – and one potentially therapeutic option – passed FDR correction in patients in the black cluster with the most identified root causal genes according to the RCS scores. In contrast, enrichment analysis identified many drugs in patients in the green cluster with the lowest RCS scores and thus relatively few root causal genes. The pink and blue clusters yielded moderate results. We conclude that drug enrichment analysis expectedly

identified more drugs for patients on the left hand side of the UMAP embedding with fewer root causal genes than on the right hand side with many simultaneous root causal genes.

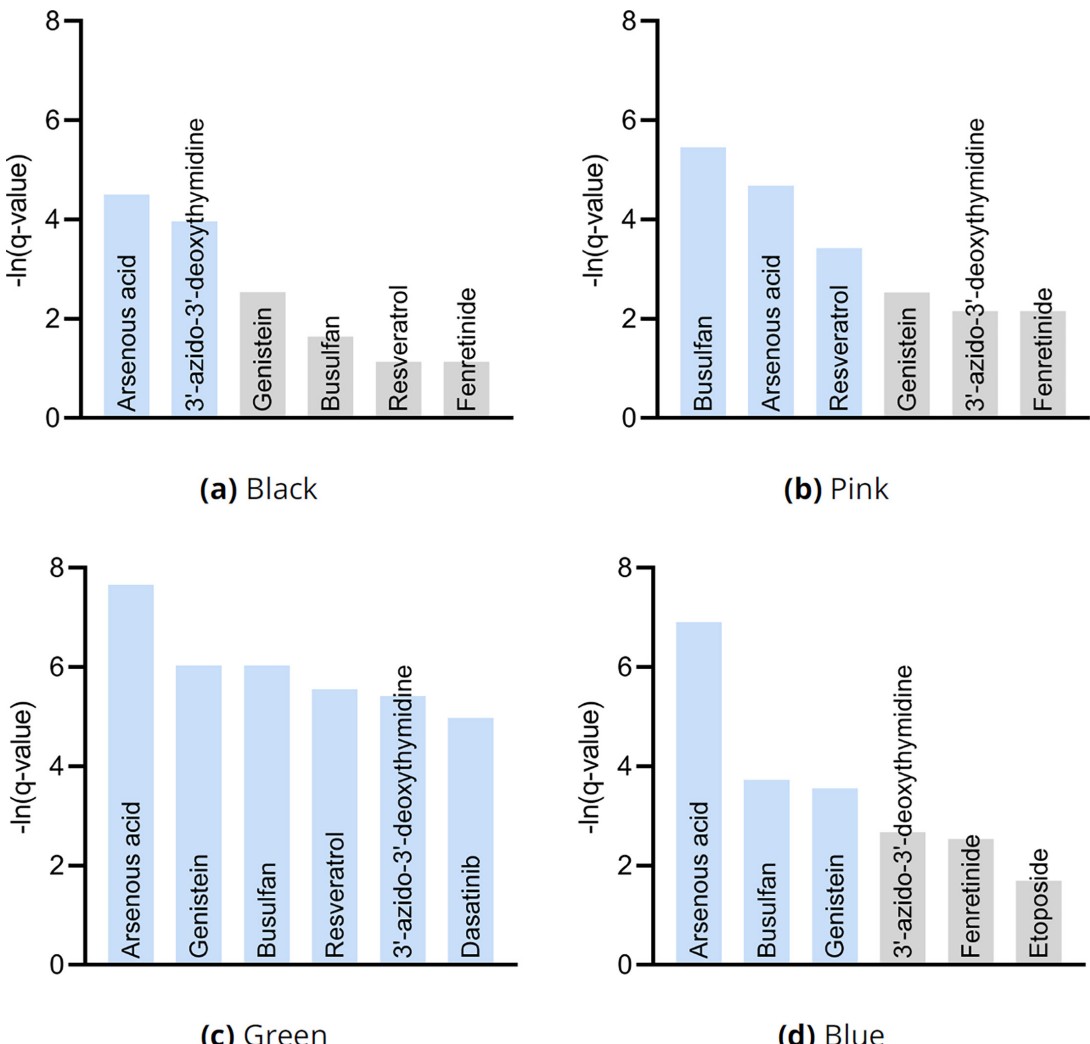

**Appendix 1—figure 10.** Drug enrichment analysis results by cluster in *Figure 3g*. The analyses recovered similar drugs across clusters, but the results for the green cluster in (**c**) were supra-significant.

## Additional results for multiple sclerosis

### Algorithm comparisons

We compared the algorithms using the MS data with the same criteria used for the AMD dataset. We summarize the results in *Appendix 1—figure 11* plotted on the next page. Only the histogram of RCSP had large probability mass centered around zero as shown in the first column. The histogram of LiNGAM contained many outliers, so it appears to spike around a value of 18. The histograms of ANM and CausalCell were again near identical. We conclude that only the histogram of RCSP supported an omnigenic root causal model in MS.

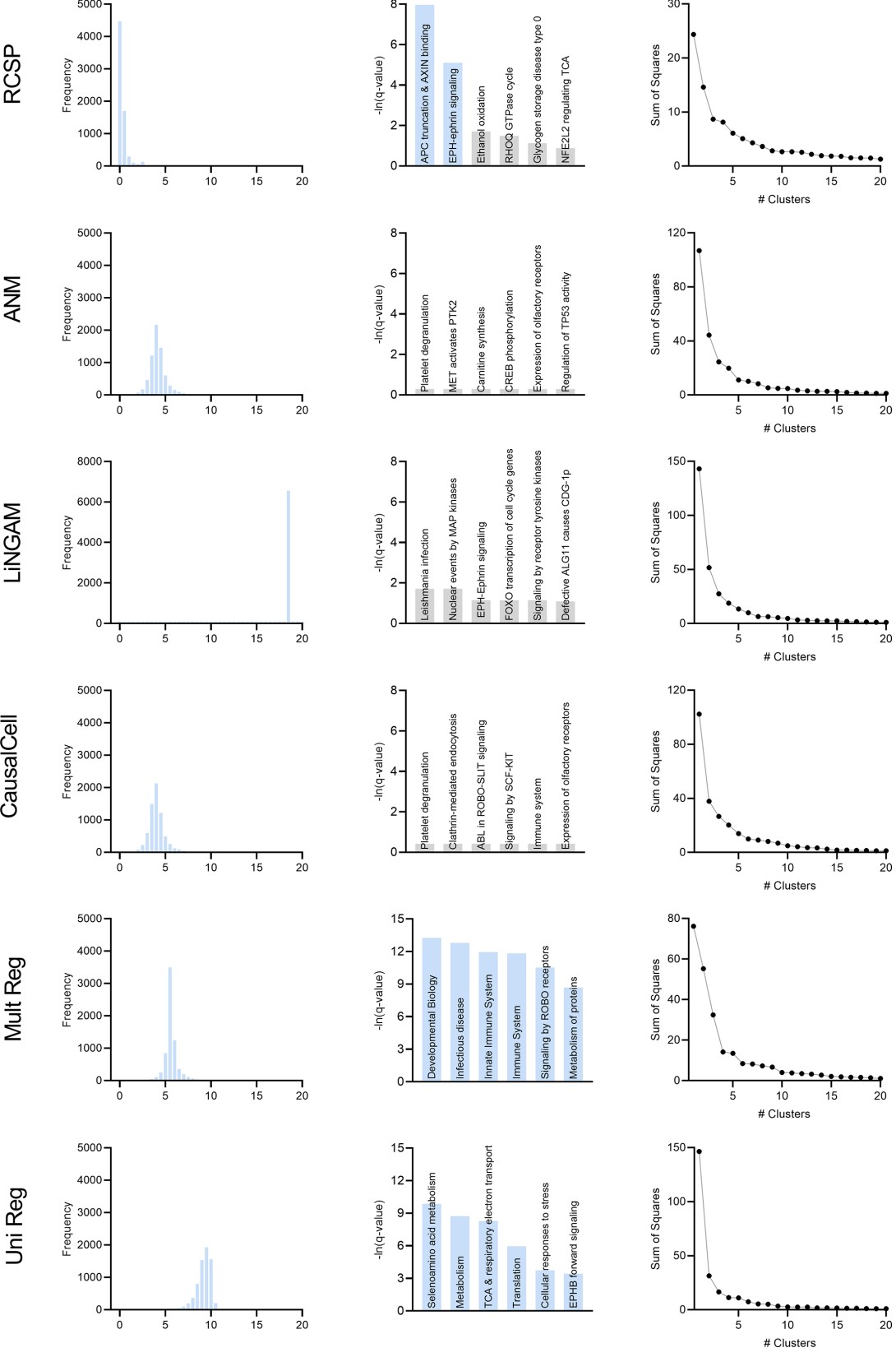

**Appendix 1—figure 11.** Comparison of the algorithms in multiple sclerosis.

We performed pathway enrichment analysis on the algorithm outputs and summarize the results in the second column of *Appendix 1—figure 11*. The functional causal models ANM, LiNGAM and

CausalCell did not identify significant pathways at an FDR corrected threshold of 0.05. In contrast, multivariate and univariate regression both identified many significant pathways in blue with no specific link to the blood brain barrier. The top six significant pathways for multivariate and univariate regression involved 112–831 and 18–545 leading edge genes, respectively. In contrast, the two significant pathways of RCSP involved only 2 and 9 leading genes. We conclude that RCSP detected pathogenic pathways of MS with the sparsest set of leading edge genes.

We finally clustered the algorithm outputs into patient subgroups. We list the sum of squares plots in the third column of *Appendix 1—figure 11*. Univariate regression did not differentiate between the patients because it detected one dominating cluster. RCSP and multivariate regression identified clear subgroups according to the elbow method, whereas the sum of squares plots for ANM, LiNGAM and CausalCell showed no clear cutoffs. We conclude that only RCSP and multivariate regression identified clear patient subgroups in MS.

In summary, only RCSP simultaneously detected an omnigenic root causal model, identified pathogenic pathways with high specificity and discovered clear patient subgroups. We therefore conclude that RCSP also outperformed all other algorithms in the MS dataset.

## Effect of sequencing depth

We plot sequencing depth against the D-RCS scores of each gene similar to the AMD dataset. We again observed a small negative correlation ($\rho$ =-0.136, *P*<2.2E-16), indicating that genes with low sequencing depths had slightly higher D-RCS scores on average (*Appendix 1—figure 12*). However, genes with the largest D-RCS scores again had a variety of sequencing depths. We conclude that sequencing depth has minimal correlation with the largest D-RCS scores.

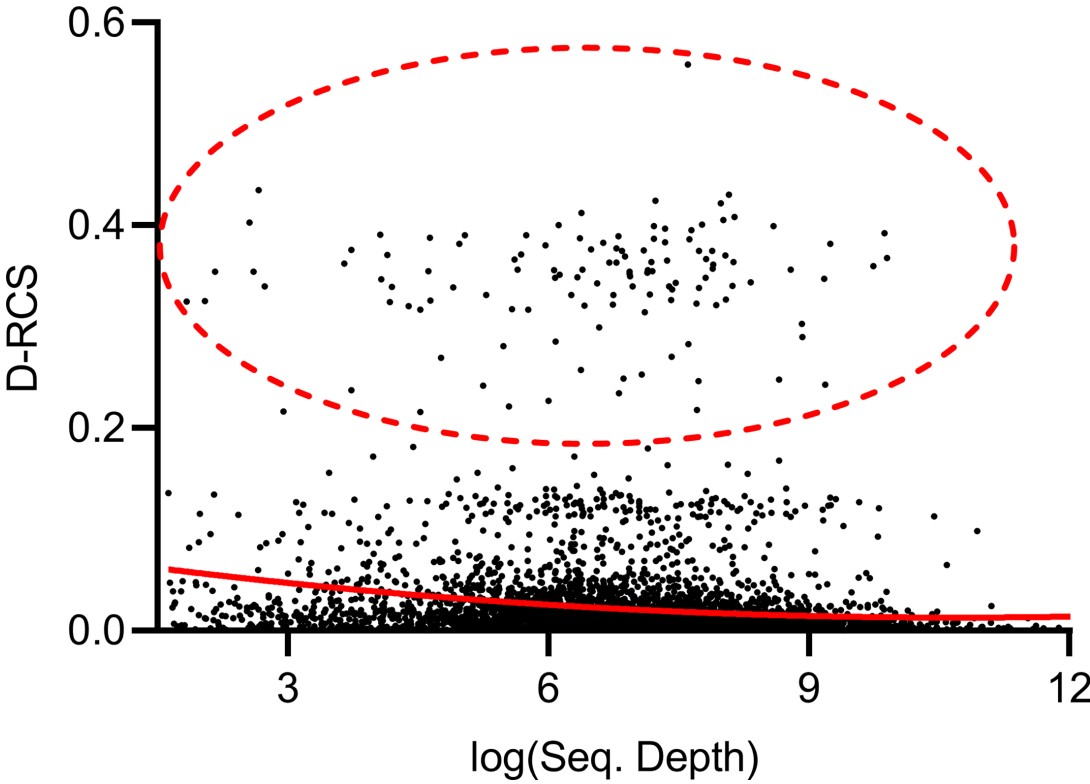

**Appendix 1—figure 12.** Mean sequencing depth of each gene plotted against their D-RCS scores in MS. Genes with the largest D-RCS scores (red ellipse) again had a variety of sequencing depths.

## Biological results

We provide the full global pathway enrichment analysis results for MS in *Appendix 1—table 2*. Pathway enrichment analysis of the individual clusters in *Figure 4f* consistently implicated EPH-ephrin signaling among the top two pathways. However, each cluster also involved one separate

additional pathway (*Appendix 1—figure 13*). The green cluster involved the same APC-AXIN pathway as the global analysis via beta-catenin. On the other hand, the blue cluster involved 'platelet sensitization by LDL.' Low density lipoprotein enhances platelet aggregation. Platelet degranulation in turn drives the generation of autoreactive T cells in the peripheral circulation during disturbance of the blood brain barrier (*Orian et al., 2021*). Finally, CTLA4 regulates T-cell homeostasis and inhibits autommunity for the pink cluster (*Basile et al., 2022*). The D-RCS scores of each cluster thus implicate different mechanisms of T cell pathology.

**Appendix 1—table 2.** Full pathway enrichment analysis results for all patients in the MS dataset. We again list up to the top three leading edge genes in the right-most column.

| Pathway | p-value | q-value | Effect Size | Leading Edge |
|---|---|---|---|---|
| APC truncation mutants have impaired AXIN binding | 1.91e-06 | 3.45E-04 | 0.960 | 55,255,527 |
| EPH-ephrin signaling | 4.23e-05 | 6.12E-03 | 0.826 | 88,741,028,976 |
| Ethanol oxidation | 2.02e-03 | 0.182 | 0.967 | 219,128 |
| RHOQ GTPase cycle | 2.72e-03 | 0.226 | 0.793 | 9,322,887,410,395 |
| Glycogen storage disease type 0 (muscle GYS1) | 2.72e-03 | 0.322 | 0.996 | 2992 |
| NFE2L2 regulating TCA cycle genes | 6.31e-03 | 0.414 | 0.970 | 41,993,417 |
| C6 deamination of adenosine | 7.42e-03 | 0.414 | 0.981 | 103,104 |
| Ion channel transport | 7.63e-03 | 0.414 | 0.728 | 5,719,854,055,515 |
| Synthesis of IP3 and IP4 in the cytosol | 7.63e-03 | 0.414 | 0.904 | 363,380,523,236 |
| Diseases associated with glycosaminoglycan metabolism | 8.21e-03 | 0.414 | 0.894 | 2,132,112,853,339 |
| Signaling by SCF-KIT | 8.67e-03 | 0.414 | 0.794 | 700,655,783,815 |

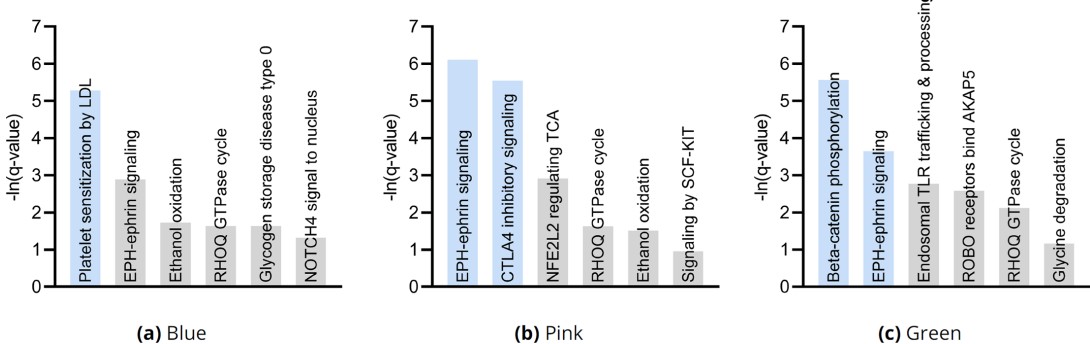

**(a)** Blue **(b)** Pink **(c)** Green

**Appendix 1—figure 13.** Pathway enrichment analysis results by cluster consistently revealed EPH-ephrin signaling as well as an additional pathway implicating T cell pathology.

The severity of MS, as assessed by the Expanded Disability Status Scale (EDSS) score, did not correlate with either dimension of the UMAP embedding (*Appendix 1—figure 14a*). The top genes in *Figure 4d* such as MNT and CERCAM also did not correlate. However, lower ranked genes such as TRIP10 did (*Appendix 1—figure 14b*). An expanded correlation analysis with the top 30 genes revealed significant correlations across a variety of lower ranked genes (*Appendix 1—figure 14c and d*). We conclude that the distribution of lower ranked genes govern the structure of the UMAP embedding in *Figure 4f*.

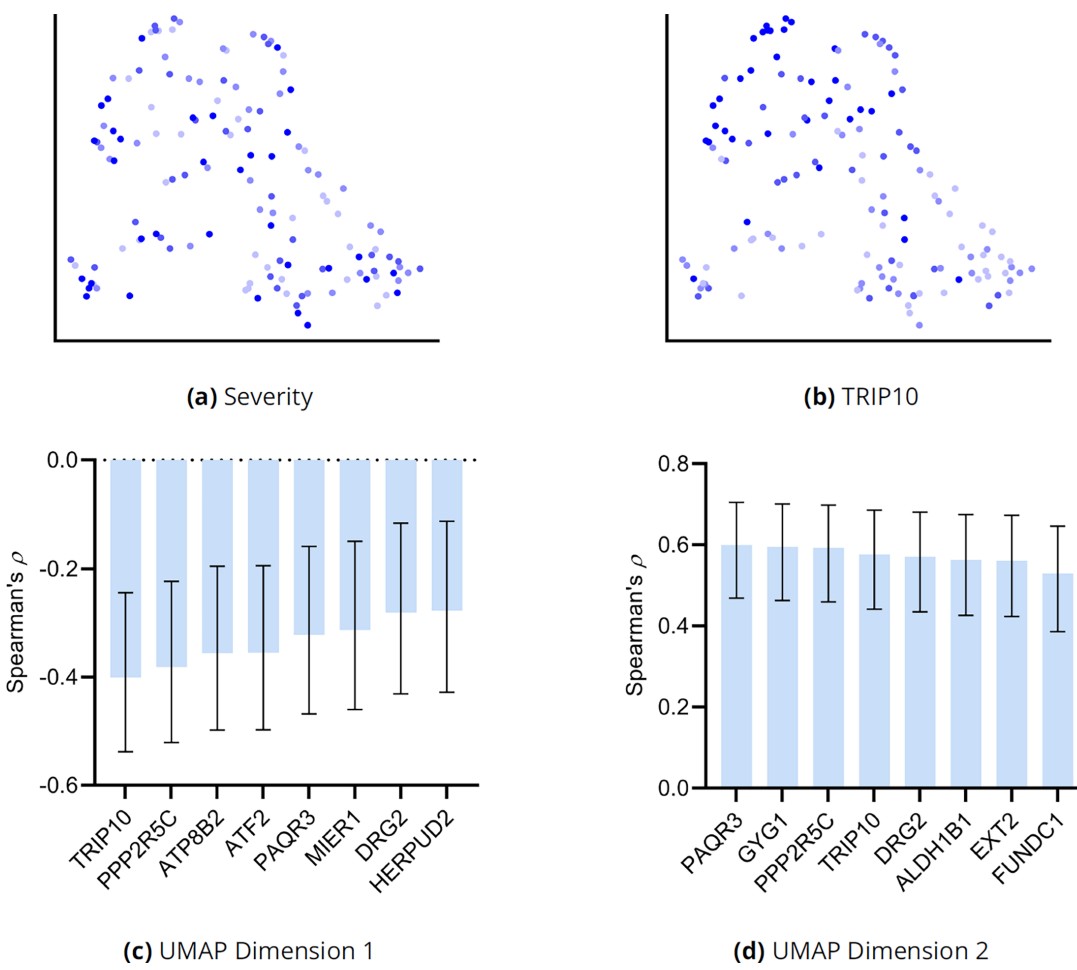

**(a)** Severity

**(b)** TRIP10

**(c)** UMAP Dimension 1

**(d)** UMAP Dimension 2

**Appendix 1—figure 14.** Additional analyses of the UMAP embedding for MS. (**a**) The UMAP dimensions did not correlate with MS severity as assessed by EDSS. However, lower ranked genes such as TRIP10 correlated with both dimensions in (**b**). We expanded the analysis to the top 30 genes and plot the genes with the highest correlations to UMAP dimension one and two in (**c**) and (**d**), respectively. Error bars denote 95% confidence intervals, and the sample size corresponded to 137 individuals.

## Proofs

Lemma 1. Assume Lipschitz continuity of the conditional expectation for all $N \geq n_0$:

$$\mathbb{E}\left|\mathbb{E}(Y|\widetilde{U}) - \mathbb{E}(Y|U, L, B)\right| \leq \mathbb{E} C_N \left|\widetilde{U} - \frac{U}{dL}\right|, \tag{6}$$

where $d = \frac{\pi_{UB}}{\sum_{\widetilde{X}_i \in \widetilde{A}} \widetilde{x}_i \pi_{iB}}$, $C_N \in O(1)$ is a positive constant, and we have taken an outer expectation on both sides. Then $\mathbb{E}(Y|\widetilde{U}) = \lim_{N \to \infty} \mathbb{E}(Y|U, L, B)$ almost surely.

*Proof.*

We can write the following sequence:

$$\mathbb{E}\left|\mathbb{E}(Y|\widetilde{U}) - \lim_{N \to \infty} \mathbb{E}(Y|U, L, B)\right| = \mathbb{E} \lim_{N \to \infty} \left|\mathbb{E}(Y|\widetilde{U}) - \mathbb{E}(Y|U, L, B)\right|$$
$$\leq \mathbb{E} \lim_{N \to \infty} C_N \left|\widetilde{U} - \frac{U}{dL}\right| \leq C\mathbb{E}\left|\widetilde{U} - \frac{1}{d}\lim_{N \to \infty}\frac{U}{L}\right| = C\mathbb{E}\left|\widetilde{U} - \frac{1}{d}\widetilde{U}d\right| = 0, \tag{7}$$

where we have applied Expression (7) at the first inequality. We have $C_N \leq C$ for all $N \geq n_0$ in the second inequality because $C_N \in O(1)$. With the above bound, choose $a > 0$ and invoke the Markov inequality:

$$\mathbb{P}\left(\left|\mathbb{E}(Y|\widetilde{U}) - \lim_{N\to\infty}\mathbb{E}(Y|U,L,B)\right| \geq a\right) \leq \frac{1}{a}\mathbb{E}\left|\mathbb{E}(Y|\widetilde{U}) - \lim_{N\to\infty}\mathbb{E}(Y|U,L,B)\right| = 0.$$

The conclusion follows because we chose $a$ arbitrarily. ∎

Proposition 1. If $E_i \not\perp\!\!\!\perp Y$ or $E_i \not\perp\!\!\!\perp Y|\mathrm{Pa}(\widetilde{X}_i)$ (or both), then $E_i$ is a root cause of $Y$.

*Proof.*

If $E_i \not\perp\!\!\!\perp Y$ or $E_i \not\perp\!\!\!\perp Y|\mathrm{Pa}(\widetilde{X}_i)$ (or both), then $E_i$ and $Y$ are d-connected by the global Markov property. Since $E_i$ is a root vertex, the d-connection implies that there exists a directed path from $E_i$ to $Y$. ∎

Proposition 2. We have $\mathbb{P}(Y|E_i, \mathrm{Pa}(\widetilde{X}_i)) = \mathbb{P}(Y|\widetilde{X}_i, \mathrm{Pa}(\widetilde{X}_i))$ under *Equation 3*.

*Proof.*

We can write:

$$\mathbb{P}(Y|E_i, \mathrm{Pa}(\widetilde{X}_i)) = \mathbb{E}_{\widetilde{X}_i|E_i, \mathrm{Pa}(\widetilde{X}_i)}\mathbb{P}(Y|E_i, \widetilde{X}_i, \mathrm{Pa}(\widetilde{X}_i)) = \mathbb{P}(Y|E_i, \widetilde{X}_i, \mathrm{Pa}(\widetilde{X}_i)) = \mathbb{P}(Y|\widetilde{X}_i, \mathrm{Pa}(\widetilde{X}_i)).$$

The second equality follows because $\widetilde{X}_i$ is a constant given $E_i$ and $\mathrm{Pa}(\widetilde{X}_i)$. The third equality follows by the global Markov property because $Y$ is a terminal vertex. ∎

Theorem 2 (Fisher consistency) Consider the same assumption as Lemma 1. If unconditional d-separation faithfulness holds, then RCSP recovers $\Phi$ almost surely as $N \to \infty$.

*Proof.* If $X_k \not\perp\!\!\!\perp P_i$ in Line 2 of Algorithm 1, then $X_k$ is a descendant of the root vertex $P_i$ under the global Markov property. Similarly, if $X_k$ is a descendant of $P_i$, then $X_k$ is d-connected to $P_i$ so $X_k \not\perp\!\!\!\perp P_i$ by unconditional d-separation faithfulness. Hence, $\mathrm{SD}(\widetilde{X}_i)$ contains only and all the surrogate descendants of $\widetilde{X}_i$ for each $\widetilde{X}_i \in \widetilde{X}$. This in turn implies that $\mathrm{SA}(\widetilde{X}_i)$ in Line 5 of Algorithm 1 contains only and all the surrogate ancestors of $\widetilde{X}_i$. Hence, RCSP now has access to the correct set $\mathrm{SA}(\widetilde{X}_i)$ as well as $B$ for each $\widetilde{X}_i \in \widetilde{X}$. We finally invoke Theorem 1 to conclude that RCSP recovers $\Phi$ almost surely as $N \to \infty$. ∎

