## [Editor Report · eLife Assessment]

This work provides an **important** framework for understanding the primary causes of disease. While the theoretical results rely on strong assumptions about the underlying causal mechanisms, the authors provide **solid** empirical evidence that the framework is robust to modest violations of these assumptions.

---

## [Referee Report · Reviewer #1 (Public review)]

Summary:

This manuscript seeks to estimate the causal effect of genes on disease. To do so, they introduce a novel algorithm, termed the Root Causal Strength using Perturbations (RCSP) algorithm. RCSP uses perturb-seq to first estimate the gene regulatory network structure among genes, and then uses bulk RNA-seq with phenotype data on the samples to estimate causal effects of genes on the phenotype conditional on the learned network structure. The authors assess the performance of RCSP in comparison to other methods via simulation. Next, they apply RCSP to two real human datasets: 513 individuals age-related macular degeneration and 137 individuals with multiple sclerosis.

Strengths:

The authors tackle an important and ambitious problem - the identification of causal contributors to disease in the context of a causal inference framework. As the authors point out, observational RNA-seq data is insufficient for this kind of causal discovery, since it is very challenging to recover the true underlying graph from observational data; interventional data are needed. However, little perturb-seq data has been generated with annotated phenotype data, and much bulk RNA-seq data has already been generated, so it is useful to propose an algorithm to integrate the two as the authors have done.

The authors also offer substantial theoretical exposition for their work, bringing to bear both the literature on causal discovery as well as literature on the genetic architecture of complex traits. They also benchmark RCSP under multiple challenging simulation settings, including an analysis of RCSP when the underlying graph is not a DAG.

Weaknesses:

The notion of a "root" causal gene - which the authors define based on a graph theoretic notion of topologically sorting graphs - requires a graph that is directed and acyclic. It is the latter that constitutes an important weakness here - it simply is a large simplification of human biology to draw out a DAG including hundreds of genes and a phenotype Y and to claim that the true graph contains no cycles. For example - consider the authors' analysis of T cell infiltration in multiple sclerosis (MS). CD4+ effector T cells have the interesting property that they are stimulated by IL2 as a growth factor; yet IL2 also stimulates the activation of (suppressive) regulatory T cells. What does it mean to analyze CD4+ regulation in disease with a graph that does not consider IL2 (or other cytokine) mediated feedback loops/cycles? To the authors' credit, in the supplementary materials they do consider a simulated example with a cyclic underling causal graph, finding that RCSP performed well comparison to an implementation of the additive noise model (ANM), LiNGAM, CausalCell, and two simpler approaches based on linear regression.

I also encourage the authors to consider more carefully when graph structure learned from perturb-seq can be ported over to bulk RNA-seq. Consider again the MS CD4+ example - the authors first start with a large perturb-seq experiment (Replogle et al., 2022) performed in K562 cells. To what extent are K562 cells, which are derived from a leukemia cell line, suitable for learning the regulatory structure of CD4+ cells from individuals with an MS diagnosis? Presumably this structure is not exactly correct - to what extent is the RCSP algorithm sensitive to false edges in this graph? The authors perform an analysis of this scenario in Supplementary Figure 4, which shows that RCSP is robust to some degree of departure from the underlying true structure. And although challenging - it would be ideal for the RCSP to model or reflect the challenges in correctly identifying the regulatory structure.

It should also be noted that in most perturb-seq experiments, the entire genome is not perturbed, and frequently important TFs (that presumably are very far "upstream" and thus candidate "root" causal genes) are not expressed highly enough to be detected with scRNA-seq. In that context - perhaps slightly modifying the language regarding RCSP's capabilities might be helpful for the manuscript - perhaps it would be better to describe it has an algorithm for causal discovery among a set of genes that were perturbed and measured, rather than a truly complete search for causal factors. Perhaps more broadly - it would also benefit the manuscript to devote slightly more text to describing the kinds of scenarios where RCSP (and similar ideas) would be most appropriately applied - perhaps a well-powered, phenotype annotated perturb-seq dataset performed in a disease relevant primary cell.

---

## [Referee Report · Reviewer #2 (Public review)]

Summary:

This paper presents a very interesting use of a causal graph framework to identify the "root genes" of a disease phenotype. Root genes are the genes that cause a cascade of events that ultimately leads to the disease phenotype, assuming the disease progression is linear.

Strengths:

- The methodology has a solid theoretical background.

- This is a novel use of the causal graph framework to infer root causes in a graph

Comments on revisions:

The authors addressed all of my comments.

---

## [Author Response]

The following is the authors’ response to the original reviews.

**Reviewer 1:**
(1) The notion of a “root” causal gene - which the authors define based on a graph theoretic notion of topologically sorting graphs - requires a graph that is directed and acyclic. It is the latter that constitutes an important weakness here - it simply is a large simplification of human biology to draw out a DAG including hundreds of genes and a phenotype *Y* and to claim that the true graph contains no cycles.

We agree that real causal graphs in biology often contain cycles. We now include additional experimental results with cyclic directed graphs in the Supplementary Materials. RCSP outperformed the other algorithms even in this setting, but we caution the reader that the theoretical interpretation of the RCS score may not coincide with a root causal effect when cycles exist:

“We also evaluated the algorithms on directed graphs with cycles. We generated a linear SEM over *ρ* + 1 = 1000 variables in X~∪y. We sampled the coefficient matrix *β* from a Bernoulli (1/(*p* − 1)) distribution but did not restrict the non-zero coefficients to the upper triangular portion of the matrix. We then proceeded to permute the variable ordering and weight each entry as in the Methods for the DAG. We repeated this procedure 30 times and report the results in Supplementary Figure 3.

RCSP again outperformed all other algorithms even in the cyclic case. The results suggest that conditioning on the surrogate ancestors also estimates the RCS well even in the cyclic case. However, we caution that an error term *Ei* can affect the ancestors of X¯ when cycles exist. As a result, the RCS may not isolate the causal effect of the error term and thus not truly coincide with the notion of a root causal effect in cyclic causal graphs.”

(2) I also encourage the authors to consider more carefully when graph structure learned from Perturb-seq can be ported over to bulk RNA-seq. Presumably this structure is not exactly correct - to what extent is the RCSP algorithm sensitive to false edges in this graph? This leap - from cell line to primary human cells - is also not modeled in the simulation. Although challenging - it would be ideal for the RCSP to model or reflect the challenges in correctly identifying the regulatory structure.

We now include additional experimental results, where we gradually increased the incongruence between the DAG modeling the Perturb-seq and the DAG modeling the bulk RNA-seq using a mixture of graphs. The performance of RCSP degraded gradually, rather than abruptly, with increasing incongruence. We therefore conclude that RCSP is robust to differences between the causal graphs representing Perturb-seq and bulk RNA-seq:

“We next assessed the performance of RCSP when the DAG underlying the Perturb-seq data differs from the DAG underlying the bulk RNA-seq data. We considered a mixture of two random DAGs in bulk RNA-seq, where one of the DAGs coincided with the Perturb-seq DAG and second alternate DAG did not. We instantiated and simulated samples from each DAG as per the previous subsection. We generated 0%, 25%, 50%, 75%, and 100% of the bulk RNA-seq samples from the alternate DAG, and the rest from the Perturb-seq DAG. We ideally would like to see the performance of RCSP degrade gracefully, as opposed to abruptly, as the percent of samples derived from the alternate DAG increases.

We summarize results in Supplementary Figure 4. As expected, RCSP performed the best when we drew all samples from the same underlying DAG for Perturb-seq and bulk RNA-seq. However, the performance of RCSP also degraded slowly as the percent of samples increased from the alternate DAG. We conclude that RCSP can accommodate some differences between the underlying DAGs in Perturb-seq and bulk RNA-seq with only a mild degradation in performance.”

(3) It should also be noted that in most Perturb-seq experiments, the entire genome is not perturbed, and frequently important TFs (that presumably are very far “upstream” and thus candidate “root” causal genes) are not expressed highly enough to be detected with scRNA-seq. In that context - perhaps slightly modifying the language regarding RCSP’s capabilities might be helpful for the manuscript - perhaps it would be better to describe it as an algorithm for causal discovery among a set of genes that were perturbed and measured, rather than a truly complete search for causal factors. Perhaps more broadly it would also benefit the manuscript to devote slightly more text to describing the kinds of scenarios where RCSP (and similar ideas) would be most appropriately applied - perhaps a well-powered, phenotype annotated Perturb-seq dataset performed in a disease relevant primary cell.

We now clarify that Perturb-seq can only identify root causal genes among the perturbed set of genes in the Discussion:

“Modern genome-wide Perturb-seq datasets also adequately perturb and measure only a few thousand, rather than all, gene expression levels. RCSP can only identify root causal genes within this perturbed and measured subset.”

We now also describe the scenario where RCSP can identify root causal genes well in the Introduction:

“Experiments demonstrate marked improvements in performance, when investigators have access to a large bulk RNA-seq dataset and a genome-wide Perturb-seq dataset from a cell line of a disease-relevant tissue.”

**Reviewer 2:**
(1) The process from health-to-disease is not linear most of the time with many checks along the way that aim to prevent the disease phenotype. This leads to a non-deterministic nature of the path from health-to-disease. In other words, with the same root gene perturbations, and depending on other factors outside of gene expression, someone may develop a phenotype in a year, another in 10 years and someone else never. Claiming that this information is included in the error terms might not be sufficient to address this issue. The authors should discuss this limitation.

The proposed approach accommodates the above non-deterministic nature. The error terms of X¯ model factors that are outside of gene expression. We model the relation from gene expression to *Y* as probabilistic rather than deterministic because Y=fY(Pa(Y),EY), where *EY* introduces stochasticity. Thus, two individuals with the same instantiations of the root causes may develop disease differently. We now clarify this in Methods:

“The error terms model root causes that are outside of gene expression, such as genetic variation or environmental factors. Moreover, the relation from gene expression to *Y* is stochastic because Y=fY(Pa(Y),EY), where *EY* introduces the stochasticity. Two individuals may therefore have the exact same error term values over X¯ but different instantiations of Y.”

(2) The paper assumes that the network connectivity will remain the same after perturbation. This is not always true due to backup mechanisms in the cells. For example, suppose that a cell wants to create product *P* and it can do it through two alternative paths: Path #1: *A* → *B* → *P*, Path #2: *A* → *C* → *P*. Now suppose that path #1 is more efficient, so when *B* can be produced, path #2 is inactive. Once the perturbation blocks element *B* from being produced, the graph connectivity changes by activation of path #2. I did not see the authors taking this into consideration, which seems to be a major limitation in using Perturb-seq results to infer conductivities.

We agree that backup mechanisms can exist and therefore now include additional experimental results, where we gradually increased the incongruence between the DAG modeling the Perturb-seq and the DAG modeling the bulk RNA-seq using a mixture of graphs. The performance of RCSP degraded gradually, rather than abruptly, with increasing incongruence. We therefore conclude that RCSP is robust to differences between the causal graphs representing Perturb-seq and bulk RNA-seq:

“We next assessed the performance of RCSP when the DAG underlying the Perturb-seq data differs from the DAG underlying the bulk RNA-seq data. We considered a mixture of two random DAGs in bulk RNA-seq, where one of the DAGs coincided with the Perturb-seq DAG and second alternate DAG did not. We generated 0%, 25%, 50%, 75%, and 100% of the bulk RNA-seq samples from the alternate DAG, and the rest from the Perturb-seq DAG. We ideally would like to see the performance of RCSP degrade gracefully, as opposed to abruptly, as the percent of samples derived from the alternate DAG increases.

We summarize results in Supplementary Figure 4. As expected, RCSP performed the best when we drew all samples from the same underlying DAG for Perturb-seq and bulk RNA-seq. However, the performance of RCSP also degraded slowly as the percent of samples increased from the alternate DAG. We conclude that RCSP can accommodate some differences between the underlying DAGs in Perturb-seq and bulk RNA-seq with only a mild degradation in performance.”

(3) There is substantial system heterogeneity that may cause the same phenotype. This goes beyond the authors claim that although the initial gene causes of a disease may differ from person to person, at some point they will all converge to changes in the same set of “root genes.” This is not true for many diseases, which are defined based on symptoms and lab tests at the patient level. You may have two completely different molecular pathologies that lead to the development of the same symptoms and test results. Breast cancer with its subtypes is a prime example of that. In theory, this issue could be addressed if there is infinite sample size. However, this assumption is largely violated in all existing biological datasets.

The proposed method accommodates the above heterogeneity. We do not assume that the root causes affect the same set of root causal genes. Instead the root causes *and* root causal genes may vary from person to person. We write in the Introduction:

“The problem is further complicated by the existence of complex disease, where a patient may have multiple root causal genes that differ from other patients even within the same diagnostic category... We thus also seek to identify *patient-specific* root causal genes in order to classify patients into meaningful biological subgroups each hopefully dictated by only a small group of genes.”

The root causal genes may further affect different downstream genes at the patient-specific level. However root causal genes tend to have many downstream effects so that virtually every gene expression level becomes correlated with *Y*. We now clarify this by describing the omnigenic root causal model in the Introduction as follows:

“Finally, application of the algorithm to two complex diseases with disparate pathogeneses recovers an *omnigenic root causal model*, where a small set of root causal genes drive pathogenesis but impact many downstream genes within each patient. As a result, nearly all gene expression levels are correlated with the diagnosis at the population level.”

(4) Were the values of the synthetic variables Z-scored?

Yes, all variables were z-scored. We now clarify this in Methods:

“We also standardized all variables before running the regressions to prevent gaming of the marginal variances in causal discovery (Reisach et al., 2021; Ng et al., 2024).”

(5) The algorithm seems to require both RNA-seq and Perturb-seq data (Algorithm 1, page 14). Can it function with RNA-seq data only? What will be different in this case?

The algorithm cannot function with observational bulk RNA-seq data only. We included Perturb-seq because causal discovery with observational RNA-seq data alone tends to be inaccurate and unstable, as highlighted by the results of CausalCell. We further emphasize that we do not rely on d-separation faithfulness in Methods, which is typically required for causal discovery from observational data alone:

“We can also claim the backward direction under d-separation faithfulness. We however avoid making this additional assumption because real biological data may not arise from distributions obeying d-separation faithfulness in practice.”

(6) Synthetic data generation: how many different graphs (SEMs) did they start from? (30?) How many samples per graph? Did they test different sample sizes?

We now clarify that we generate 30 random SEMs, each associated with a DAG. We used 200 samples for the bulk RNA-seq to mimic a relatively large but common sample size. We also drew 200 samples for each perturbation or control in the Perturb-seq data. We did not consider multiple sample sizes due to the time required to complete each run. Instead, we focused on a typical scenario where investigators would apply RCSP. We now write the following in the Methods:

“We drew 200 samples for the bulk RNA-seq data to mimic a large but common dataset size. We introduced knockdown perturbations in Perturb-seq by subtracting an offset of two in the softplus function: X¯i←softplus⁡(X¯i−2). We finally drew 200 samples for the control and each perturbation condition to generate the Perturb-seq data. We repeated the above procedure 30 times.” We also include the following in Results:

“We obtained 200 cell samples from each perturbation, and another 200 controls without perturbations. We therefore generated a total of 2501 × 200 = 500,200 single cell samples for each Perturb-seq dataset. We simulated 200 bulk RNA-seq samples.”

(7) The presentation of comparative results (Supplementary Figures 4 and 7) is not clear. No details are given on how these results were generated. (what does it mean “The first column denotes the standard deviation of the outputs for each algorithm?”) Why all other methods have higher SD differences than RCSP? Is it a matter of scaling? Shouldn’t they have at least some values near zero since the authors “added the minimum value so that all histograms begin at zero?”

Each of these supplementary figures contains a 6 by 3 table of figures. By the first column, we mean column one (with rows 1 through 6) of each figure. The D-RCS and D-SD scores represent standard deviations of the RCS and SD scores from zero of each gene, respectively. We can similarly compute the standard deviation of the outputs of the algorithms. We now clarify this in the Supplementary Materials:

“The figure contains 6 rows and 3 columns. Similar to the D-RCS, we can compute the standard deviation of the output of each algorithm from zero for each gene. The first column in Supplementary Figure 7 denotes the histograms of these standard deviations across the genes.”

Many histograms do not appear to start at zero because the bars are too small to be visible. We now clarify this in the Supplementary Materials as well:

“Note that the bars at zero are not visible for many algorithms, since only a few genes attained standard deviations near the minimum.”

(8) Why RCSP results are more like a negative binomial distribution and every other is kind of normal?

All other methods have higher standard deviations than RCSP because they fail to compute an accurate measure of the root causal effect. Recall that, just like a machine has a few root causal problems, only a few root casual genes have large root causal effects under the omnigenic root causal model. The results of RCSP look more like a negative binomial distribution because most RCS scores are concentrated around zero and only a few RCS scores are large – consistent with the omnigenic root causal model. The other algorithms fail to properly control for the upstream genes and thus attain large standard deviations for nearly all genes. We now clarify these points in the Supplementary Materials as follows:

“If an algorithm accurately identifies root causal genes, then it should only identify a few genes with large conditional root causal effects under the omnigenic root causal model. The RCSP algorithm had a histogram with large probability mass centered around zero with a long tail to the right. The standard deviations of the outputs of the other algorithms attained large values for nearly all genes. Incorporating feature selection and causal discovery with CausalCell introduced more outliers in the histogram of ANM. We conclude that only RCSP detected an omnigenic root causal model.”

(9) What is the significance of genes changing expression “from left to right” in a UMAP plot? (e.g., Fig. 3h and 3g)

The first UMAP dimension captured the variability of the RCS scores for most root causal genes. As a result, we could focus our analysis on the black cluster in Figure 3 (g) with large RCS scores in the subsequent pathway enrichment analysis summarized in Figure 3 (j). If two dimensions were involved, then we would need to analyze at least two clusters (e.g., black and pink), but this was not the case. We now clarify this in Results:

“The RCS scores of most of the top genes exhibited a clear gradation increasing only from the left to the right hand side of the UMAP embedding; we plot an example in Figure 3 (h). We found three exceptions to this rule among the top 30 genes (example in Figure 3 (i) and see Supplementary Materials). RCSP thus detected genes with large RCS scores primarily in the black cluster of Figure 3 (g). Pathway enrichment analysis within this cluster alone yielded supra-significant results on the same pathway detected in the global analysis...”

(10) The authors somewhat overstate the novelty of their algorithm. Representation of GRNs as causal graphs dates back in 2000 with the work of Nir Friedman in yeast. Other methods were developed more recently that look on regulatory network changes at the single sample level which the authors do not seem to be aware (e.g., Ellington et al, NeurIPS 2023 workshop GenBio and Bushur et al, 2019, Bioinformatics are two such examples). The methods they mention are for single cell data and they are not designed to connect single sample-level changes to a person’s phenotype. The RCS method needs to be put in the right background context in order to bring up what is really novel about it.

We agree that many methods already exist for uncovering associational, predictive (Markov, neighborhood) and causal gene regulatory networks. We now cite the above papers. However, the novelty in our manuscript is not causal graph discovery, but rather estimation of root causal effects, detection of root causal genes, and the proposal of the omnigenic root causal model. We now clarify this in the

Introduction:

“Many algorithms focus on discovering associational or predictive relations, sometimes visually represented as gene regulatory networks (Costa et al., 2017; Ellington et al., 2023). Other methods even identify causal relations (Friedman et al., 2000; Wang et al., 2023; Wen et al., 2000; Buschur et al., 2000), but none pinpoint the *first* gene expression levels that ultimately generate the vast majority of pathogenesis. Simply learning a causal graph does not resolve the issue because causal graphs do not summarize the effects of *unobserved* root causes, such as unmeasured environmental changes or variants, that are needed to identify all root causal genes. We therefore define the Root Causal Strength (RCS) score...”

**Reviewer 3:**
(1) Several assumptions of the method are problematic. The most concerning is that the observational expression changes are all causally upstream of disease. There is work using Mendelian randomization (MR) showing that the *opposite* is more likely to be true: most differential expression in disease cohorts is a consequence rather than a cause of disease (Porcu et al., 2021). Indeed, the oxidative stress of AMD has known cellular responses including the upregulation of p53. The authors need to think carefully about how this impacts their framework. Can the theory say anything in this light? Simulations could also be designed to address robustness.

Strictly speaking, we believe that differential expression in disease most likely has a cyclic causal structure: gene expression causes a diagnosis or symptom severity, and a diagnosis or symptom severity lead to treatments and other behavioral changes that perturb gene expression. For example, revTMWR in Porcu et al. (2021) uses trans-variants that are less likely to *directly* cause gene expression and instead directly cause a phenotype. However, TWMR as proposed in Porcu et al. (2019) instead uses cis-eQTLs and finds many putative causal relations from gene expression to phenotype. Thus, both causal directions likely hold.

RCSP uses disease-relevant tissue believed to harbor gene expression levels that cause disease. However, RCSP theoretically cannot handle the scenario where *Y* is a non-sink vertex *and* is a parent of a gene expression level because modern Perturb-seq datasets usually do not perturb or measure *Y*. We therefore empirically investigated the degree of error by running experiments, where we set *Y* to a non-sink vertex, so that it can cause gene expression. We find that the performance of RCSP degrades considerably for gene expression levels that contain *Y* as a parent. Thus RCSP is sensitive to violations of the sink target assumption:

“We finally considered the scenario where *Y* is a non-sink (or non-terminal) vertex. If *Y* is a parent of a gene expression level, then we cannot properly condition on the parents because modern Perturbseq datasets usually do not intervene on *Y* or measure *Y* . We therefore empirically investigated the degradation in performance resulting from a non-sink target *Y*, in particular for gene expression levels where *Y* is a parent. We again simulated 200 samples from bulk RNA-seq and each condition of Perturbseq with a DAG over 1000 vertices, an expected neighborhood size of 2 and a non-sink target *Y* . We then removed the outgoing edges from *Y* and resampled the DAG with a sink target. We compare the results of RCSP for both DAGs in gene expression levels where *Y* is a parent. We plot the results in Supplementary Figure 5. As expected, we observe a degradation in performance when *Y* is not terminal, where the mean RMSE increased from 0.045 to 0.342. We conclude that RCSP is sensitive to violations of the sink target assumption.”

(2) A closely related issue is the DAG assumption of no cycles. This assumption is brought to bear because it is required for much classical causal machinery, but is unrealistic in biology where feedback is pervasive. How robust is RCSP to (mild) violations of this assumption? Simulations would be a straightforward way to address this.

We agree that real causal graphs in biology often contain cycles. We now include additional experimental results with cyclic directed graphs in the Supplementary Materials. RCSP outperformed the other algorithms even in this setting, but we caution the reader that the theoretical interpretation of the RCS score may not coincide with a root causal effect when cycles exist:

“We also evaluated the algorithms on directed graphs with cycles. We generated a linear SEM over *p* + 1 = 1000 variables in X~∪y. We sampled the coefficient matrix *β* from a Bernoulli (1/(*p* − 1)) distribution but did not restrict the non-zero coefficients to the upper triangular portion of the matrix. We then proceeded to permute the variable ordering and weight each entry as in the Methods for the DAG. We repeated this procedure 30 times and report the results in Supplementary Figure 3.

RCSP again outperformed all other algorithms even in the cyclic case. The results suggest that conditioning on the surrogate ancestors also estimates the RCS well even in the cyclic case. However, we caution that an error term *Ei* can affect the ancestors of X¯, when cycles exist. As a result, the RCS may not isolate the causal effect of the error term and thus not truly coincide with the notion of a root causal effect in cyclic causal graphs.”

(3) The authors spend considerable effort arguing that technical sampling noise in *X* can effectively be ignored (at least in bulk). While the mathematical arguments here are reasonable, they miss the bigger picture point that the measured gene expression *X* can only ever be a noisy/biased proxy for the expression changes that caused disease: (1) Those events happened before the disease manifested, possibly early in development for some conditions like neurodevelopmental disorders. (2) bulk RNA-seq gives only an average across cell-types, whereas specific cell-types are likely “causal.” (3) only a small sample, at a single time point, is typically available. Expression in other parts of the tissue and at different times will be variable.

We agree that many other sources of error exist. The causal model of RNA-expression in Methods corresponds to a single snapshot in time for each sample. We now clarify this in the Methods as follows:

“We represent a snapshot of a biological causal process using an SEM over X~∪y obeying Equation (3).”

We thus only detect the root causal genes in a single snapshot in time for each sample in bulk RNA-seq. If we cannot detect the root causal effect in a gene due to the signal washing out over time as in (1), or if the root causal effect in different cell types cancel each other out to exactly zero in bulk as in (2), then we cannot detect those root causal genes even with an infinite sample size.

(4) While there are connections to the omnigenic model, the latter is somewhat misrepresented. The authors refer to the “core genes” of the omnigenic model as being at the end (longitudinal) of pathogenesis. The omnigenic model makes no statements about temporal ordering: in causal inference terminology the core genes are simply the direct causes of disease.

We now clarify that we use the word *pathogenesis* to mean the causal cascade from root causes to the diagnosis. In this case, the direct causes of the diagnosis correspond to the end of pathogenesis, while the root causes correspond to the beginning. For example, if X1→X2→Y, with *Y* a diagnosis, then *X*_1_ is a root causal gene while *X*_2_ is a core (direct causal) gene. We now clarify this in the Introduction:

“*Root causes of disease* correspond to the most upstream causes of a diagnosis with strong causal effects on the diagnosis. *Pathogenesis* refers to the causal cascade from root causes to the diagnosis. Genetic and non-genetic factors may act as root causes and affect gene expression as an intermediate step during pathogenesis. We introduce root causal gene expression levels – or *root causal genes* for short – that correspond to the initial changes to *gene expression* induced by genetic and non-genetic root causes that have large causal effects on a downstream diagnosis (Figure 1 (a)). Root causal genes differ from core genes that directly cause the diagnosis and thus lie at the end, rather than at the beginning, of pathogenesis (Boyle et al., 2017).”

(5) A key observation underlying the omnigenic model is that genetic heritability is spread throughout the genome (and somewhat concentrated near genes expressed in disease relevant cell types). This implies that (almost) all expressed genes, or their associated (e)SNPs, are “root causes”.

We now clarify that genetic heritability can be spread throughout the genome in the omnigenic root causal model as well in the Discussion:

“Further, each causal genetic variant tends to have only a small effect on disease risk in complex disease because the variant can directly cause *Y* or directly cause any causal gene including those with small root causal effects on *Y* ; thus, all error terms that cause *Y* can model genetic effects on *Y*. However, the root causal model further elaborates that genetic *and non-genetic factors* often combine to produce a few root causal genes with large root causal effects, where non-genetic factors typically account for the majority of the large effects in complex disease. Many variants may therefore cause many genes in diseases with only a few root causal genes.”

We finally add Figure 5 into the Discussion as a concrete example illustrating the omnigenic root causal model:

(6) The claim that root causal genes would be good therapeutic targets feels unfounded. If these are highly variable across individuals then the choice of treatment becomes challenging. By contrast the causal effects may converge on core genes before impacting disease, so that intervening on the core genes might be preferable. The jury is still out on these questions, so the claim should at least be made hypothetical.

We clarify that we do not claim that root causal genes are better treatment targets than core genes in terms of magnitudes of causal effects on the phenotype. For example, in the common cold with a virus as the root cause, giving a patient an antiviral will eliminate fever and congestion, but so will giving a decongestant and an antipyretic. We only claim that treating root causal genes can eliminate disease near its pathogenic onset, just like giving an antiviral can eliminate the viral load and stop pathogenesis. We write the following the Introduction:

“Treating root causal genes can modify disease pathogenesis in its entirety, whereas targeting other causes may only provide symptomatic relief... Identifying root causal genes is therefore critical for developing treatments that eliminate disease near its pathogenic onset.”

We also further clarify in the Discussion that root causal genes account for deleterious causal effects not captured by the diagnosis *Y*:

“We finally emphasize that the root causal model accounts for all deleterious effects of the root causal genes, whereas the core gene model only captures the deleterious effects captured by the diagnosis *Y*. For example, the *disease* of diabetes causes retinopathy, but retinopathy is not a part of the diagnostic criteria of diabetes. As a result, the gene expression levels that cause retinopathy but not the *diagnosis* of diabetes are not core genes, even though they are affected by the root causal genes.”

We do agree that root causal genes may differ substantially between patients, although it is unclear if the heterogeneity is too great to develop treatments.

(7) The closest thing to a gold standard I believe we have for “root causal genes” is integration of molecular QTLs and GWAS, specifically coloc/MR. Here the “E” of RCSP are explicitly represented as SNPs. I don’t know if there is good data for AMD but there certainly is for MS. The authors should assess the overlap with their results. Another orthogonal avenue would be to check whether the root causal genes change early in disease progression.

Colocalization and Mendelian randomization unfortunately cannot identify root causal effects because they all attempt, either heuristically (colocalization) or rigorously (MR), to identify variants that cause each gene expression level rather than variants that *directly* cause each gene expression level and thus make up the error terms. We therefore need new methods that can identify direct causal variants in order to assess overlap.

We checked whether root causal genes change early in disease progression using knowledge of pathogenesis. In particular, oxidative stress induces pathogenesis in AMD, and RCSP identified root causal genes involved in oxidative stress in AMD:

“The pathogenesis of AMD involves the loss of RPE cells. The RPE absorbs light in the back of the retina, but the combination of light and oxygen induces oxidative stress, and then a cascade of events such as immune cell activation, cellular senescence, drusen accumulation, neovascularization and ultimately fibrosis (Barouch et al., 2007). We therefore expect the root causal genes of AMD to include genes involved in oxidative stress during early pathogenesis. The gene MIPEP with the highest D-RCS score in Figure 3 (d) indeed promotes the maturation of oxidative phosphorylation-related proteins (Shi et al., 2011). The second gene SLC7A5 is a solute carrier that activates mTORC1 whose hyperactivation increases oxidative stress via lipid peroxidation (Nachef et al., 2021; Go et al., 2020). The gene HEATR1 is involved in ribosome biogenesis that is downregulated by oxidative stress (Turi et al., 2018). The top genes discovered by RCSP thus identify pathways known to be involved in oxidative stress.”

Similarly, T cell infiltration across the blood brain barrier initiates pathogenesis in MS, and RCSP identified root causal genes involved in this infiltration:

“Genes with the highest D-RCS scores included MNT, CERCAM and HERPUD2 (Figure 4 (d)). MNT is a MYC antagonist that modulates the proliferative and pro-survival signals of T cells after engagement of the T cell receptor (Gnanaprakasam et al., 2017). Similarly, CERCAM is an adhesion molecule expressed at high levels in microvessels of the brain that increases leukocyte transmigration across the blood brain barrier (Starzyk et al., 2000). HERPUD2 is involved in the endoplasmic-reticulum associated degradation of unfolded proteins (Kokame et al., 2000). Genes with the highest D-RCS scores thus serve key roles in known pathogenic pathways of MS.”

(8) The available Perturb-seq datasets have limitations beyond on the control of the authors. (1) The set of genes that are perturbed. The authors address this by simply sub-setting their analysis to the intersection of genes represented in the perturbation and observational data. However, this may mean that a true ancestor of X is not modeled/perturbed, limiting the formal claims that can be made. Additionally, some proportion of genes that are nominally perturbed show little to no actual perturbation effect (for example, due to poor guide RNA choice) which will also lead to missing ancestors.

We now clarify that Perturb-seq can only identify root causal genes among the adequately perturbed set of genes in the Discussion:

“Modern genome-wide Perturb-seq datasets also only adequately perturb and measure a few thousand, rather than all, gene expression levels. RCSP can only identify root causal genes within this perturbed and measured subset.”

(9) The authors provide no mechanism for statistical inference/significance for their results at either the individual or aggregated level. While I am a proponent of using effect sizes more than p-values, there is still value in understanding how much signal is present relative to a reasonable null.

We now explain that RCSP does not perform statistical inference in Methods because it is not clear how to define the appropriate cut-off for the RCS score under the null distribution:

“We focus on statistical estimation rather than statistical inference because Φ*i >* 0 when *Ei* causes *Y* under mild conditions, so we reject the null hypothesis that Φ*i* = 0 for many genes if many gene expression levels cause *Y*. However, just like a machine typically breaks down due to only one or a few root causal problems, we hypothesize that only a few genes have large RCS scores Φ*i* ≫ 0 even in complex disease.”

(10) I agree with the authors that age coming out of a “root cause” is potentially encouraging. However, it is also quite different in nature to expression, including being “measured” exactly. Will RCSP be biased towards variables that have lower measurement error?

We tested the above hypothesis by plotting sequencing depth against the D-RCS scores of each gene. We observed a small negative correlation between sequencing depth and D-RCS scores, indicating the D-RCS scores are slightly biased upwards with low sequencing depth. However, genes with the largest D-RCS scores exhibited a wide variety of sequencing depths in both MS and AMD, suggesting that sequencing depth has minimal effect on the largest D-RCS scores. We now explain these results for AMD in the Supplementary Materials:

“Theorem 1 states that RCS scores may exhibit bias with insufficient sequencing depth. The genes with large D-RCS scores may therefore simply have low sequencing depths. To test this hypothesis, we plotted sequencing depth against D-RCS scores. Consistent with Theorem 1, we observed a small negative correlation between D-RCS and sequencing depth (*ρ* = −0.16, p=2.04E-13), and D-RCS scores exhibited greater variability at the lowest sequencing depths (Supplementary Figure 8). However, genes with the largest D-RCS scores had mean sequencing depths interspersed between 20 and 3000. We conclude that genes with the largest D-RCS scores had a variety of sequencing depths ranging from low to high.”

We also report the results for MS:

“We plot sequencing depth against the D-RCS scores of each gene similar to the AMD dataset. We again observed a small negative correlation (*ρ* = −0.136, p_<_2.2E-16), indicating that genes with low sequencing depths had slightly higher D-RCS scores on average (Supplementary Figure 12). However, genes with the largest D-RCS scores again had a variety of sequencing depths. We conclude that sequencing depth has minimal correlation with the largest D-RCS scores.”

(11) Finally, it’s a stretch to call K562 cells “lymphoblasts.” They are more myeloid than lymphoid.

We now clarify that K562 cells are undifferentiated blast cells that can be induced to differentiate into lymphoblasts in Results:

“We next ran RCSP on 137 samples collected from CD4+ T cells of multiple sclerosis (MS; GSE137143) as well as Perturb-seq data of 1,989,578 undifferentiated blast cells that can be induced to differentiate into lymphoblasts, or the precursors of T cells and other lymphocytes.”